# Online Label Shift: Optimal Dynamic Regret meets Practical Algorithms

**Dheeraj Baby**[*]
UC Santa Barbara
dheeraj@ucsb.edu

**Saurabh Garg**[*]
Carnegie Mellon University
sgarg2@andrew.cmu.edu

**Tzu-Ching Yen**[*]
Carnegie Mellon University
tzuchiny@andrew.cmu.edu

**Sivaraman Balakrishnan**
Carnegie Mellon University
sbalakri@andrew.cmu.edu

**Zachary C. Lipton**
Carnegie Mellon University
zlipton@andrew.cmu.edu

**Yu-Xiang Wang**
UC Santa Barbara
yuxiangw@cs.ucsb.edu

## Abstract

This paper focuses on supervised and unsupervised online label shift, where the class marginals $Q(y)$ varies but the class-conditionals $Q(x|y)$ remain invariant. In the unsupervised setting, our goal is to adapt a learner, trained on some offline labeled data, to changing label distributions given unlabeled online data. In the supervised setting, we must both learn a classifier and adapt to the dynamically evolving class marginals given only labeled online data. We develop novel algorithms that reduce the adaptation problem to online regression and guarantee optimal dynamic regret without any prior knowledge of the extent of drift in the label distribution. Our solution is based on bootstrapping the estimates of *online regression oracles* that track the drifting proportions. Experiments across numerous simulated and real-world online label shift scenarios demonstrate the superior performance of our proposed approaches, often achieving 1-3% improvement in accuracy while being sample and computationally efficient. Code is publicly available at this url.

## 1 Introduction

Supervised machine learning algorithms are typically developed assuming independent and identically distributed (iid) data. However, real-world environments evolve dynamically [55, 67, 46, 29]. Absent further assumptions on the nature of the shift, such problems are intractable. One line of research has explored causal structures such as covariate shift [64], label shift [61, 51], and missingness shift [86], for which the optimal target predictor is identified from labeled source and unlabeled target data. Let's denote the feature-label pair of an example by $(x, y)$. Label shift addresses the setting where the label marginal distribution $Q(y)$ may change but the conditional distribution $Q(x|y)$ remains fixed. Most prior work addresses the batch setting for unsupervised adaptation, where a single shift occurs between a source and target population [61, 51, 1, 2, 1, 26]. However, in the real world, shifts are more likely to occur continually and unpredictably, with data arriving in an *online* fashion. A nascent line of research tackles online distribution shift, typically in settings where labeled data is available in real time [4], seeking to minimize the *dynamic regret*.

Researchers have only begun to explore the role that structures like label shift might play in such online settings. Initial attempts to learn under unsupervised online label shifts were made by Wu et al. [76] and Bai et al. [8], both of which rely on reductions to Online Convex Optimization (OCO) [37, 54]. This line of research aims in updating a classification model based on online data so that the overall regret is controlled. However, Wu et al. [76] only control for *static regret* against a fixed

---

[*]Equal Contribution

37th Conference on Neural Information Processing Systems (NeurIPS 2023).

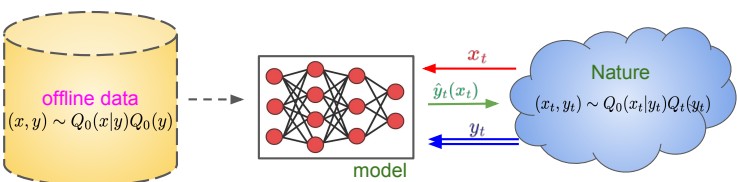

Figure 1: *UOLS and SOLS setup.* Dashed (double) arrows are exclusive to UOLS (SOLS) settings. Other objects are common to both setups. Central question: how to adapt the model in real-time to drifting label marginals based on all the available data so far?

classifier (or model) in hindsight and makes the assumption of the convexity (of losses), which is often violated in practice. In the face of online label shift, where the class marginals can vary across rounds, a more fitting notion is to control the *dynamic regret* against a sequence of models in hindsight. Motivated by this observation, Bai et al. [8] control for the dynamic regret. However, their approach is based on updating model parameters (of the classifier) with online gradient descent and relying on convex losses limits the applicability of their methods (e.g. algorithms in Bai et al. [8] can not be employed with decision tree classifiers).

In this paper, we study the problem of learning classifiers under Online Label Shift (OLS) in both *supervised* and *unsupervised* settings (Fig.1). In both these settings, the distribution shifts are an online process that respects the label shift assumption. Our primary goal is to develop algorithms that side-step convexity assumptions and at the same time *optimally* adapt to the non-stationarity in the label drift. In the Unsupervised Online Label Shift (UOLS) problem, the learner is provided with a pool of labeled offline data sampled iid from the distribution $Q_0(x,y)$ to train an initial model $f_0$. Afterwards, at every online round $t$, few *unlabeled* data points sampled from $Q_t(x)$ are presented. The goal is to adapt $f_0$ to the non-stationary target distributions $Q_t(x,y)$ so that we can accurately classify the unlabelled data. By contrast, in Supervised Online Label Shift (SOLS), our goal is to learn classifiers from *only* the (labeled) samples that arrive in an online fashion from $Q_t(x,y)$ at each time step, while simultaneously adapting to the non-stationarity induced due to changing label proportions. While SOLS is similar to online learning under non-stationarity, UOLS differs from classical online learning as the test label is not seen during online adaptation. Below are the list of contributions of this paper.

- **Unsupervised adaptation.** For the UOLS problem, we provide a reduction to online regression (see Defn. 1), and develop algorithms for adapting the initial classifier $f_0$ in a computationally efficient way leading to *minimax optimal* dynamic regret. Our approach achieves the best-of-both worlds of Wu et al. [76], Bai et al. [8] by controlling the dynamic regret while allowing us to use expressive black-box models for classification (Sec. 3).

- **Supervised adaptation.** We develop algorithms for SOLS problem that lead to *minimax optimal* dynamic regret without assuming convexity of losses (Sec. 4). Our theoretically optimal solution is based on weighted Empirical Risk Minimization (wERM) with weights tracked by online regression. Motivated by our theory, we also propose a *simple continual learning* baseline which achieves empirical performance competitive to the wERM from scratch at each time step across several semi-synthetic SOLS problems while being $15\times$ more efficient in computation cost.

- **Low switching regressors.** We propose a black-box reduction method to convert an optimal online regression algorithm into another algorithm that switches decisions *sparingly* while *maintaining minimax optimality*. This method is relevant for online change point detection. We demonstrate its application in developing SOLS algorithms to train models only when significant distribution drift is detected, while maintaining statistical optimality (App. D and Algorithm 6).

- **Extensive empirical study.** We corroborate our theoretical findings with experiments across numerous simulated and real-world OLS scenarios spanning vision and language datasets (Sec. 5). Our proposed algorithms often improve over the best alternatives in terms of both final accuracy and label marginal estimation. This advantage is particularly prominent with limited initial holdout data (in the UOLS problem) highlighting the *sample efficiency* of our approach.

**Notes on technical novelties.** Even-though online regression is a well studied technique, to the best of our knowledge, it is not used before to address the problem of online label shift. It is precisely the usage of regression which lead to tractable adaptation algorithms while side-stepping convexity assumptions thereby allowing us to use very flexible models for classification. This is in stark contrast

to OCO based reductions in [76] and [8]. We propose new theoretical frameworks and identify the right set of assumptions for materializing the reduction to online regression. It was not evident initially that this link would lead to *minimax optimal* dynamic regret rates as well as *consistent* empirical improvement over prior works. Proof of the lower bounds requires adapting the ideas from non-stationary stochastic optimization [10] in a non-trivial manner. Further, none of the proposed methods require the prior knowledge of the extent of distribution drift.

## 2 Problem Setup

Let $\mathcal{X} \subseteq \mathbb{R}^d$ be the input space and $\mathcal{Y} = [K] := \{1, 2, \ldots, K\}$ be the output space. Let $Q$ be a distribution over $\mathcal{X} \times \mathcal{Y}$ and let $q(\cdot)$ denotes the corresponding label marginal. $\Delta_K$ is the $K$-dimensional simplex. For a vector $v \in \mathbb{R}^K$, $v[i]$ is its $i^{th}$ coordinate. We assume that we have a hypothesis class $\mathcal{H}$. For a function $f \in \mathcal{H} : \mathcal{X} \to \Delta_K$, we also use $f(i|x)$ to indicate $f(x)[i]$. With $\ell(f(x), y)$, we denote the loss of making a prediction with the classifier $f$ on $(x, y)$. $L$ denotes the expected loss, i.e., $L = \mathbb{E}_{(x,y) \sim Q}[\ell(f(x), y)]$. $\tilde{O}(\cdot)$ hides dependencies in absolute constants and poly-logarithmic factors of horizon and failure probabilities.

In this work, we study online learning under distribution shift, where the distribution $Q_t(x, y)$ may continuously change with time. Throughout the paper, we focus on the *label shift* assumption where the distribution over label proportions $q_t(y)$ can change arbitrarily but the distribution of the covariate conditioned on a label value (i.e., $Q_t(x|y)$) is assumed to be invariant across all time steps. We refer to this setting as Online Label Shift (OLS). Here, we consider settings of unsupervised and supervised OLS settings captured in Frameworks 1 and 3 respectively. In both settings, at round $t$ a sample $(x_t, y_t)$ is drawn from a distribution with density $Q_t(x_t, y_t)$. In the UOLS setting, the label is not revealed to the learner. However, we assume access to offline labeled data sampled iid from $Q_0$ which we use to train an initial classifier $f_0$. The goal is to adapt the initial classifier $f_0$ to drifting label distributions. In contrast, for the SOLS setting, the label is revealed to the learner after making a prediction and the goal is to learn a classifier $f_t \in \mathcal{H}$ for each time step.

Next, we formally define the concept of online regression which will be central to our discussions. Simply put, an online regression algorithm tracks a ground truth sequence from noisy observations.

**Definition 1** (online regression). *Fix any $T > 0$. The following interaction scheme is defined to be the online regression protocol.*

- *At round $t \in [T]$, an algorithm predicts $\hat{\theta}_t \in \mathbb{R}^K$.*
- *A noisy version of ground truth $z_t = \theta_t + \epsilon_t$ is revealed where $\theta_t, \epsilon_t \in \mathbb{R}^K$, and $\|\epsilon_t\|_2, \|\theta_t\|_2 \leq B$. Further the noise $\epsilon_t$ are independent across time with $E[\epsilon_t] = 0$ and $\mathrm{Var}(\epsilon_t[i]) \leq \sigma^2 \ \forall i \in [K]$.*

*An online regression algorithm aims to control $\sum_{t=1}^T \|\hat{\theta}_t - \theta_t\|_2^2$. Moreover, the regression algorithm is defined to be adaptively minimax optimal if with probability at least $1 - \delta$, $\sum_{t=1}^n \|\hat{\theta}_t - \theta_t\|_2^2 = \tilde{O}(T^{1/3} V_T^{2/3})$ without knowing $V_T$ ahead of time. Here $V_T := \sum_{t=2}^T \|\theta_t - \theta_{t-1}\|_1$ is termed as the Total Variation (TV) of the sequence $\theta_{1:T}$.*

## 3 Unsupervised Online Label Shift

In this section, we develop a framework for handling the UOLS problem. We summarize the setup in Framework 1. Since in practice, we may need to work with classifiers such as deep neural networks or decision trees, we do not impose convexity assumptions on the (population) loss of the classifier as a function of the model parameters. Despite the absence of such simplifying assumptions, we provide performance guarantees for our label shift adaption techniques so that they are certified to be fail-safe.

Under the label shift assumption, we have $Q_t(y|x)$ as a re-weighted version of $Q_0(y|x)$:

$$Q_t(y|x) = \frac{Q_t(y)}{Q_t(x)} Q_t(x|y) = \frac{Q_t(y)}{Q_t(x)} Q_0(x|y) = \frac{Q_t(y)Q_0(x)}{Q_t(x)Q_0(y)} Q_0(y|x) \propto \frac{Q_t(y)}{Q_0(y)} Q_0(y|x),$$

where the second equality is due to the label shift assumption. Hence, a reasonable strategy is to re-weight the initial classifier $f_0$ with label proportions (estimate) at the current step, since we only have to correct the label distribution shift. This re-weighting technique is widely used for offline label shift correction [51, 2, 1] and for learning under label imbalance [41, 74, 18].

**Framework 1** Unsupervised Online Label Shift (UOLS) protocol

**Input**: Initial classifier $f_0 : \mathcal{X} \to \Delta_K$ trained on offline labeled dataset $\{(x_i, y_i)\}_{i=1}^{N}$ sampled iid from $Q_0$;

1: $f_1 = f_0$
2: **for** each round $t \in [T]$ **do**
3:    Nature samples $x_t \in \mathcal{X}$ and $y_t \in \mathcal{Y}$, with $(x_t, y_t) \sim Q_t$; Only $x_t$ is revealed to the learner.
4:    Learner predicts a label $i \sim f_t(x_t) \in \Delta_K$.
5:    $f_{t+1} = \mathcal{A}(f_0, x_{1:t})$, where $\mathcal{A}$ is strategy to adapt the classifier based on past data.
6: **end for**

**Algorithm 2** `RegressAndReweight` to handle UOLS

**Input**: i) Online regression oracle ALG; ii) Initial classifier $f_0$; iii) The confusion matrix $C$; iv) The label marginal $q_0 \in \mathcal{D}$ of the training distribution;

1: At round $t$, get the classifier covariate $x_t$.
2: Let $\widehat{q}_t = \Pi_{\mathcal{D}}(\text{ALG}(s_{1:t-1}))$, where $\Pi_{\mathcal{D}}(x) = \operatorname{argmin}_{y \in \mathcal{D}} \|y - x\|_2$.
3: Sample a label $i$ with probability $\propto \frac{\widehat{q}_t(i)}{q_0(i)} f_0(i|x_t)$.
4: Let $s_t = C^{-1} f_0(x_t)$.
5: Update the online regression oracle with the estimate $s_t$.

Our starting point in developing a framework is inspired by Wu et al. [76], Bai et al. [8] . For self-containedness, we briefly recap their arguments next. We refer interested readers to their papers for more details. Wu et al. [76] considers a hypothesis class of re-weighted initial classifier $f_0$. The loss of a hypothesis is parameterised by the re-weighting vector. They use tools from OCO to optimise the loss and converge to a best fixed classifier. However as noted in Wu et al. [76], the losses are not convex with respect to the re-weight vector in practice. Hence usage of OCO techniques is not fully satisfactory in their problem formulation.

In a complementary direction, Bai et al. [8] abandons the idea of re-weighting. Instead, they update the parameters of a model at each round using online gradient descent and a loss function whose expected value is assumed to be convex with respect to model parameters. They provide dynamic regret guarantees against a sequence of changing model parameters in hindsight, and connects it to the variation of the true label marginals. More precisely, they provide algorithms with $\sum_{t=1}^{T} L_t(w_t) - L_t(w_t^*)$ to be well controlled where $w_t^*$ is the best model parameter to be used at round $t$ and $L_t$ is a (population level) loss function. However, there are some scopes for improvement in this direction as well. For example, the convexity assumption can be easily violated when working with interpretable models based on decision trees, or if we want to retrain few final layers of a deep classifier based on new data. Further as noted in the experiments (Sec. 5), their methods based on retraining the classifier require more data than re-weighting based methods. Our experiments also indicate that re-weighting can be computationally cheaper than re-training without sacrificing the classifier accuracy.

Thus, on the one hand, the work of Wu et al. [76] allows us to use the power of expressive initial classifiers while only controlling the static regret against a fixed hypothesis. On the other hand, the work of Bai et al. [8] allows controlling the dynamic regret while limiting the flexibility of deployed models. We next provide our framework for handling label shifts that achieves the best of both worlds by controlling the dynamic regret while allowing the use of expressive *blackbox* models.

In summary, we estimate the sequence of online label marginals and leverage the idea of re-weighting an initial classifier as in Wu et al. [76]. In particular, given an estimate $\widehat{q}_t(y)$ of the true label marginal at round $t$, we compute the output of the re-weighted classifier $f_t$ as $\frac{\widehat{q}_t(y)}{q_0(y)} f_0(y|x)/Z$ where $Z = \sum_y \frac{\widehat{q}_t(y)}{q_0(y)} f_0(y|x)$. However, to get around the issue of non-convexity, we separate out the process of estimating the re-weighting vectors via a reduction to online regression which is a well-defined and convex problem with computationally efficient off-the-shelf algorithms readily available. Second, and more importantly, Wu et al. [76] competes with the best *fixed* re-weighted hypothesis. However, in the problem setting of label shift, the true label marginals are in fact changing. Hence, we control the *dynamic regret* against a sequence of re-weighted hypotheses in hindsight. All proofs for the next sub-section are deferred to App. C.

### 3.1 Proposed algorithm and performance guarantees

We start by presenting our assumptions. This is followed by the main algorithm for UOLS and its performance guarantees. Similar to the treatment in Bai et al. [8], we assume the following.

**Assumption 1.** *Assume access to the true label marginals $q_0 \in \Delta_K$ of the offline training data and the true confusion matrix $C \in \mathbb{R}^{K \times K}$ with $C_{ij} = E_{x \sim Q_0(\cdot|y=j)}, f_0(i|x)$. Further the minimum singular value $\sigma_{min}(C) = \Omega(1)$ is bounded away from zero.*

As noted in prior work [51, 26], the invertibility of the confusion matrix holds whenever the classifier $f_0$ has good accuracy and the true label marginal $q_0$ assigns a non-zero probability to each label. Though we assume perfect knowledge of the label marginals of the training data and the associated confusion matrix, this restriction can be easily relaxed to their empirical counterparts computable from the training data. The finite sample error between the empirical and population quantities can be bounded by $O(1/\sqrt{N})$ where $N$ is the number of initial training data samples. To this end, we operate in the regime where the time horizon obeys $T = O(\sqrt{N})$. However, similar to Bai et al. [8], we make this assumption mainly to simplify presentation without trivializing any aspect of the OLS problem.

Next, we present our assumptions on the loss function. Let $p \in \Delta_K$. Consider a classifier that predicts a label $\widehat{y}(x)$, by sampling $\widehat{y}(x)$ according to the distribution that assigns a weight $\frac{p(i)}{q_0(i)} f_0(i|x)$ to the label $i$. Define $L_t(p)$ to be any non-negative loss that ascertains the quality of the marginal $p$. For example, $L_t(p) = E[\ell(\hat{y}(x), y)]$ where the expectation is taken wrt the randomness in the draw $(x, y) \sim Q_t$ and in sampling $\hat{y}(x)$. Here $\ell$ is any classification loss (e.g. 0-1, cross-entropy).

**Assumption 2** (Lipschitzness of loss functions). *Let $\mathcal{D}$ be a compact and convex domain. Assume that $L_t(p)$ is $G$ Lipschitz with $p \in \mathcal{D} \subseteq \Delta_K$, i.e, $L_t(p_1) - L_t(p_2) \leq G\|p_1 - p_2\|_2$ for any $p_1, p_2 \in \mathcal{D}$. The constant $G$ need not be known ahead of time.*

We show in Lemmas 11 and 12 that the above assumption is satisfied under mild regularity conditions. Furthermore, the prior works such as Wu et al. [76] and Bai et al. [8] also require that losses are Lipschitz with a *known* Lipschitz constant apriori to set the step sizes for their OGD based methods.

The main goal here is to design appropriate re-weighting estimates such that the *dynamic regret*:

$$R_{\text{dynamic}}(T) = \sum_{t=1}^{T} L_t(\hat{q}_t) - L_t(q_t) \leq \sum_{t=1}^{T} G\|\hat{q}_t - q_t\|_2 \tag{1}$$

is controlled where $\hat{q}_t \in \Delta_K$ is the estimate of the true label marginal $q_t$. Thus we have reduced the problem of handling OLS to the problem of online estimation of the true label marginals.

Under label shift, we can get an unbiased estimate of the true marginals at any round via the techniques in Lipton et al. [51], Azizzadenesheli et al. [2], Alexandari et al. [1]. More precisely, $s_t = C^{-1} f_0(x_t)$ has the property that $E[s_t] = q_t$ (see Lemma 15). Further, the variance of the estimate $s_t$ is bounded by $1/\sigma_{min}^2(C)$. Unfortunately, these unbiased estimates can not be directly used to track the moving marginals $q_t$. This is because the total squared error $\sum_{t=1}^{T} E[\|s_t - q_t\|_2^2]$ grows linearly in $T$ as the sum of the variance of the point-wise estimates accumulates unfavorably over time.

To get around these issues, one can use online regression algorithms such as FLH [38] with online averaging base learners or the Aligator algorithm [7]. These algorithms use ensemble methods to (roughly) output running averages of $s_t$ where the variation in the *true* label marginals is small enough. The averaging within intervals where the true marginals change slowly helps to reduce the overall variance while injecting only a small bias. We use such *online regression oracles* to track the moving marginals and re-calibrate the initial classifier. Overall, Algorithm 2 summarizes our method which has the following performance guarantee.

**Theorem 2.** *Suppose we run Algorithm 2 with the online regression oracle ALG as FLH-FTL (App. F) or Aligator [7]. Then under Assumptions 1 and 2, we have*

$$E[R_{dynamic}(T)] = \tilde{O}\left( \frac{K^{1/6}T^{2/3}V_T^{1/3}}{\sigma_{min}^{2/3}(C)} + \frac{\sqrt{KT}}{\sigma_{min}(C)} \right),$$

*where $V_T := \sum_{t=2}^{T}\|q_t - q_{t-1}\|_1$ and the expectation is taken with respect to randomness in the revealed co-variates. Further, this result is attained without prior knowledge of $V_T$.*

**Remark 3.** *We emphasize that any valid online regression oracle ALG can be plugged into Algorithm 2. This implies that one can even use transformer-based time series models to track the moving marginals $q_t$. Further, we have the flexibility of choosing the initial classifier to be any* black-box *model that outputs a distribution over the labels.*

**Framework 3** Supervised Online Label Shift (SOLS) protocol

**input** A hypothesis class $\mathcal{H}$.
 1: **for** each round $t \in [T]$ **do**
 2:   Nature samples $N$ iid data points $x_{t,1:N} \in \mathcal{X}$ and $y_{t,1:N} \in \mathcal{Y}$, with each $(x_{t,i}, y_{t,i}) \sim Q_t$; $x_{t,1:N}$ is revealed to the learner.
 3:   For each $i \in [N]$, learner predicts a label $f_t(x_{t,i})$.
 4:   The label $y_{t,i} \in \mathcal{Y}$ for each $i \in [N]$ is revealed.
 5:   $f_{t+1} = \mathcal{A}(f_t, \{x_{1:t,1:N}, y_{1:t,1:N}\})$ where algorithm $\mathcal{A}$ updates the classifier with past data.
 6: **end for**

**Algorithm 4** `TrainByWeights` to handle SOLS

**input** Online regression oracle ALG, hypothesis class $\mathcal{H}$
 1: At round $t \in [T]$, get estimated label marginal $\hat{q}_t$ from ALG($s_{1:t-1}$).
 2: Update the hypothesis with weighted ERM:
$$f_t = \underset{f \in \mathcal{H}}{\arg\min} \sum_{i=1}^{t-1} \sum_{j=1}^{N} \frac{\hat{q}_t(y_{i,j})}{\hat{q}_i(y_{i,j})} \ell(f(x_{i,j}), y_{i,j}) \quad (2)$$
 3: Get co-variates $x_{t,1:N}$ and make predictions with $f_t$
 4: Get labels $y_{t,1:N}$
 5: Compute $s_t[i] = \frac{1}{N} \sum_{j=1}^{N} \mathbb{I}\{y_{t,j} = i\}$ for all $i \in [K]$.
 6: Update ALG with the empirical label marginals $s_t$.

**Remark 4.** *Unlike prior works such as [76, 8], we do not need a pre-specified bound on the gradient of the losses. Consequently Eq.(1) holds for the smallest value of the Lipschitzness coefficient $G$, leading to tight regret bounds. Further, the projection step in Line 2 of Algorithm 2 is done only to safeguard our theory against pathological scenarios with unbounded Lipschitz constant for losses. In our experiments, we do not perform such projections.*

We next show that the performance guarantee in Theorem 2 is optimal (modulo factors of $\log T$) in a minimax sense.

**Theorem 5.** *Let $V_T \leq 64T$. There exists a loss function, a domain $\mathcal{D}$ (in Assumption 2), and a choice of adversarial strategy for generating the data such that for any algorithm, we have $\sum_{t=1}^{T} E([L_t(\hat{q}_t)] - L_t(q_t)) = \Omega\left(\max\{T^{2/3}V_T^{1/3}, \sqrt{T}\}\right)$, where $\hat{q}_t \in \mathcal{D}$ is the weight estimated by the algorithm and $q_t \in \mathcal{D}$ is the label marginal at round $t$ chosen by the adversary. Here the expectation is taken with respect to the randomness in the algorithm and the adversary.*

## 4 Supervised Online Label Shift

In this section, we focus on the SOLS problem where the labels are revealed to the learner after it makes decisions. Framework 3 summarizes our setup. Let $f_t^* := \arg\min_{f \in \mathcal{H}} L_t(f)$ be the population minimiser. We aim to control the *dynamic regret* against the best sequence of hypotheses in hindsight:

$$R_{\text{dynamic}}^{\mathcal{H}}(T) =: \sum_{t=1}^{T} L_t(f_t) - L_t(f_t^*). \quad (3)$$

If the SOLS problem is convex, it reduces to OCO [37, 54] and existing works provide $\tilde{O}(T^{2/3}V_T^{1/3})$ dynamic regret guarantees [82]. However, in practice, since loss functions are seldom convex with respect to model parameters in modern machine learning, the performance bounds of OCO algorithms cease to hold true. In our work, we extend the generalization guarantees of ERM from statistical learning theory [11] to the SOLS problem. All proofs of next sub-section are deferred to App. E.

### 4.1 Proposed algorithms and performance guarantees

We start by providing a simple initial algorithm whose computational complexity and flexibility will be improved later. Note that due to the label shift assumption, for any $j, t \in [T]$, we have $E_{(x,y) \sim Q_t}[\ell(f(x), y)] = E_{(x,y) \sim Q_j}\left[\frac{q_t(y)}{q_j(y)} \ell(f(x), y)\right]$. Here we assume that the true label marginals $q_t(y) > 0$ for all $t \in [T]$ and all $y \in [K]$. Based on this, we propose a simple weighted ERM approach (Algorithm 4) where we use an online regression oracle to estimate the label marginals from the (noisy) empirical label marginals computed with observed labeled data. With weighted

ERM and plug-in estimates of importance weights, we can obtain our classifier $f_t$. One can expect that by adequately choosing the online regression oracle ALG, the risk of the hypothesis $f_t$ computed will be close to that of $f_t^*$. Here the degree of closeness will also depend on the number of data points seen thus far. Consequently, Algorithm 4 controls the dynamic regret (Eq.(3)) in a graceful manner. We have the following performance guarantee:

**Theorem 6.** *Suppose the true label marginal satisfies $\min_{t,k} q_t(k) \geq \mu > 0$. Choose the online regression oracle in Algorithm 4 as FLH-FTL (App. F) or Aligator from Baby et al. [7] with its predictions clipped such that $\hat{q}_t[k] \geq \mu$. Then with probability at least $1 - \delta$, Algorithm 4 produces hypotheses with $R_{dynamic}^{\mathcal{H}} = \tilde{O}\left(T^{2/3}V_T^{1/3} + \sqrt{T \log(|\mathcal{H}|/\delta)}\right)$, where $V_T = \sum_{t=2}^{T}\|q_t - q_{t-1}\|_1$. Further, this result is attained without any prior knowledge of the variation budget $V_T$.*

The above rate contains the sum of two terms. The second term is the familiar rate seen in the supervised statistical learning theory literature under iid data [11]. The first term reflects the price we pay for adapting to distributional drift in the label marginals. While we prove this result for finite hypothesis sets, the extension to infinite sets is direct by standard covering net arguments [70].

**Remark 7.** *Theorem 6 requires that the estimates of the label marginals to be clipped from below by $\mu$. This is done only to facilitate theoretical guarantees by enforcing that the importance weights used in Eq.(2) do not become unbounded. However, note that only the labels we actually observe enters the objective in Eq.(2). In particular, if a label has very low probability of getting sampled at a round, then it is unlikely that it enters the objective. Due to this reason, in our experiments, we haven't used the clipping operation (see Section 5 and Appendix F for more details).*

The proof of the theorem uses concentration arguments to establish that the risk of the hypothesis $f_t$ is close to the risk of the optimal $f_t^*$. However, unlike the standard offline supervised setting with iid data, for any fixed hypothesis, the terms in the summation of Eq.(2) are correlated through the estimates of the online regression oracle. We handle it by introducing uncorrelated surrogate random variables and bounding the associated discrepancy. Next, we show (near) minimax optimality of the guarantee in Theorem 6.

**Theorem 8.** *Let $V_T \leq T/8$. There exists a choice of hypothesis class, loss function, and adversarial strategy of generating the data such that $R_{dynamic}^{\mathcal{H}} = \Omega\left(T^{2/3}V_T^{1/3} + \sqrt{T \log(|\mathcal{H}|)}\right)$, where the expectation is taken with respect to randomness in the algorithm and adversary.*

**Remark 9.** *Though the rates in Theorems 5 and 8 are similar, we note that the corresponding regret definitions are different. Hence the minimax rates are not directly comparable between the supervised and unsupervised settings.*

Even-though Algorithm 4 has attractive performance guarantees, it requires retraining with weighted ERM at every round. This can be computationally expensive. To alleviate this issue, we design a new online change point detection algorithm (Algorithm 5 in App. D) that can adaptively discover time intervals where the label marginals change slow enough. We show that the new online change point detection algorithm can be used to significantly reduce the number of retraining steps without sacrificing statistical efficiency (up to constants). Due to space constraints, we defer the exact details to App. D. We remark that our change point detection algorithm is applicable to general online regression problems and hence can be of independent interest to online learning community.

**Remark 10.** *Algorithm 5 helps to reduce the run-time complexity. However, both Algorithms 4 and 5 have the drawback of storing all data points accumulated over the online rounds. This is reminiscent to FTL / FTRL type algorithms from online learning. We leave the task of deriving theoretical guarantees with reduced storage complexity under non-convex losses as an important future direction.*

## 5 Experiments[2]

### 5.1 UOLS Setup and Results

**Setup** Following the dataset setup of Bai et al. [8], we conducted experiments on synthetic and common benchmark data such as MNIST [50], CIFAR-10 [49], Fashion [77], EuroSAT [40], Arxiv [15], and SHL [31, 71]. For each dataset, the original data is split into labeled data available

---

[2]Code is publicly available at `https://github.com/Anon-djiwh/OnlineLabelShift`.

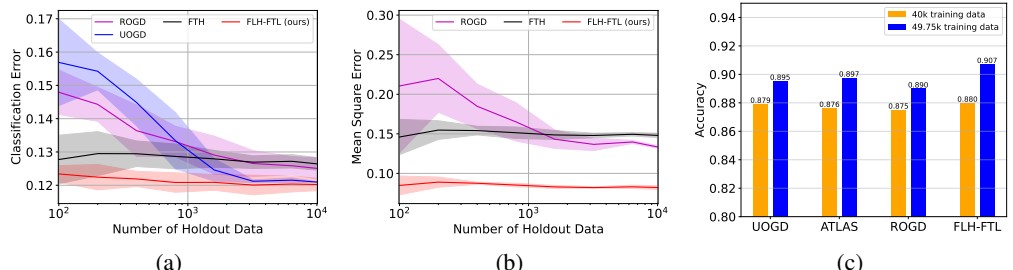

(a)               (b)               (c)

Figure 2: *Results on the UOLS problem.* **(a) and (b):** Ablation on CIFAR10 with monotone shift over sizes of holdout data used to update model parameters and compute confusion matrix, with amount of training data held fixed. FLH-FTL (ours) outperforms all other alternatives throughout in classification error and mean square error in label marginal estimation. Unlike the alternatives, the performance of FLH-FTL (ours) is unaffected by the decrease in amount of holdout data. **(c):** CIFAR10 results with monotone shift using varying amount of training data, with the remaining labeled data used as holdout (total number of samples fixed to 50k). The performance of FLH-FTL is minimally impacted by the reduction in the quantity of holdout data, thus yielding the greatest advantage from utilizing a larger volume of training data.

| Methods | Synthetic | | MNIST | | CIFAR | | EuroSAT | | Fashion | | ArXiv | |
|---|---|---|---|---|---|---|---|---|---|---|---|---|
| | Ber | Sin | Ber | Sin | Ber | Sin | Ber | Sin | Ber | Sin | Ber | Sin |
| Base | $8.6_{\pm0.2}$ | $8.2_{\pm0.3}$ | $4.9_{\pm0.4}$ | $3.9_{\pm0.0}$ | $16_{\pm0}$ | $16_{\pm0}$ | $13_{\pm0}$ | $13_{\pm0}$ | $15_{\pm0}$ | $15_{\pm0}$ | $23_{\pm1}$ | $19_{\pm0}$ |
| OFC | $6.4_{\pm0.6}$ | $5.5_{\pm0.2}$ | $4.4_{\pm0.5}$ | $3.2_{\pm0.3}$ | $12_{\pm1}$ | $11_{\pm0}$ | $11_{\pm1}$ | $10_{\pm1}$ | $7.9_{\pm0.1}$ | $7.1_{\pm0.1}$ | $20_{\pm2}$ | $15_{\pm0}$ |
| Oracle | $3.7_{\pm0.8}$ | $3.9_{\pm0.2}$ | $2.5_{\pm0.5}$ | $1.5_{\pm0.1}$ | $5.4_{\pm0.5}$ | $5.8_{\pm0.1}$ | $3.9_{\pm0.3}$ | $4.1_{\pm0.1}$ | $3.7_{\pm0.2}$ | $3.6_{\pm0.1}$ | $7.7_{\pm1.0}$ | $5.1_{\pm0.1}$ |
| FTH | $6.5_{\pm0.6}$ | $5.7_{\pm0.3}$ | $4.5_{\pm0.6}$ | $\mathbf{3.3_{\pm0.2}}$ | $11_{\pm0}$ | $\mathbf{11_{\pm0}}$ | $10_{\pm0}$ | $\mathbf{9.6_{\pm0.0}}$ | $8.5_{\pm0.3}$ | $\mathbf{6.9_{\pm0.4}}$ | $20_{\pm1}$ | $\mathbf{14_{\pm0}}$ |
| FTFWH | $6.6_{\pm0.5}$ | $5.7_{\pm0.3}$ | $4.5_{\pm0.6}$ | $\mathbf{3.3_{\pm0.2}}$ | $11_{\pm1}$ | $\mathbf{11_{\pm0}}$ | $9.8_{\pm0.4}$ | $\mathbf{9.6_{\pm0.1}}$ | $8.2_{\pm0.6}$ | $\mathbf{6.9_{\pm0.4}}$ | $20_{\pm1}$ | $\mathbf{14_{\pm0}}$ |
| ROGD | $7.9_{\pm0.3}$ | $7.2_{\pm0.6}$ | $6.2_{\pm2.8}$ | $4.4_{\pm1.5}$ | $16_{\pm3}$ | $13_{\pm0}$ | $14_{\pm1}$ | $13_{\pm1}$ | $10_{\pm1}$ | $8.2_{\pm0.7}$ | $23_{\pm2}$ | $17_{\pm1}$ |
| UOGD | $8.1_{\pm0.6}$ | $7.5_{\pm0.6}$ | $5.4_{\pm0.6}$ | $4.0_{\pm0.0}$ | $14_{\pm0}$ | $14_{\pm1}$ | $10_{\pm1}$ | $9.8_{\pm0.7}$ | $11_{\pm2}$ | $11_{\pm2}$ | $21_{\pm1}$ | $17_{\pm1}$ |
| ATLAS | $8.0_{\pm1.0}$ | $7.5_{\pm0.6}$ | $5.2_{\pm0.6}$ | $3.7_{\pm0.2}$ | $13_{\pm0}$ | $13_{\pm1}$ | $10_{\pm1}$ | $9.9_{\pm0.7}$ | $12_{\pm2}$ | $12_{\pm2}$ | $21_{\pm1}$ | $16_{\pm0}$ |
| FLH-FTL (ours) | $\mathbf{5.4_{\pm0.7}}$ | $\mathbf{5.4_{\pm0.4}}$ | $\mathbf{4.4_{\pm0.7}}$ | $\mathbf{3.3_{\pm0.2}}$ | $\mathbf{10_{\pm0}}$ | $11_{\pm0}$ | $\mathbf{9.2_{\pm0.4}}$ | $\mathbf{9.6_{\pm0.1}}$ | $\mathbf{7.7_{\pm0.4}}$ | $7.0_{\pm0.0}$ | $\mathbf{19_{\pm1}}$ | $\mathbf{14_{\pm0}}$ |

| | Synthetic | | MNIST | | CIFAR | | EuroSAT | | Fashion | | ArXiv | |
|---|---|---|---|---|---|---|---|---|---|---|---|---|
| | Ber | Sin | Ber | Sin | Ber | Sin | Ber | Sin | Ber | Sin | Ber | Sin |
| FTH | $0.19_{\pm0.01}$ | $0.10_{\pm0.00}$ | $0.27_{\pm0.00}$ | $0.14_{\pm0.00}$ | $0.27_{\pm0.01}$ | $0.14_{\pm0.00}$ | $0.27_{\pm0.00}$ | $0.14_{\pm0.00}$ | $0.29_{\pm0.01}$ | $\mathbf{0.14_{\pm0.01}}$ | $0.29_{\pm0.01}$ | $\mathbf{0.15_{\pm0.00}}$ |
| FTFWH | $0.19_{\pm0.02}$ | $0.09_{\pm0.00}$ | $0.26_{\pm0.02}$ | $0.13_{\pm0.00}$ | $0.25_{\pm0.02}$ | $0.13_{\pm0.00}$ | $0.25_{\pm0.01}$ | $\mathbf{0.13_{\pm0.00}}$ | $0.25_{\pm0.04}$ | $\mathbf{0.14_{\pm0.01}}$ | $0.27_{\pm0.02}$ | $\mathbf{0.15_{\pm0.00}}$ |
| ROGD | $0.29_{\pm0.03}$ | $0.24_{\pm0.01}$ | $0.41_{\pm0.08}$ | $0.37_{\pm0.06}$ | $0.39_{\pm0.04}$ | $0.30_{\pm0.05}$ | $0.43_{\pm0.04}$ | $0.35_{\pm0.03}$ | $0.37_{\pm0.02}$ | $0.30_{\pm0.01}$ | $0.34_{\pm0.03}$ | $0.28_{\pm0.01}$ |
| FLH-FTL (ours) | $\mathbf{0.10_{\pm0.01}}$ | $\mathbf{0.08_{\pm0.00}}$ | $\mathbf{0.15_{\pm0.01}}$ | $\mathbf{0.12_{\pm0.00}}$ | $\mathbf{0.17_{\pm0.01}}$ | $\mathbf{0.13_{\pm0.00}}$ | $\mathbf{0.16_{\pm0.01}}$ | $\mathbf{0.13_{\pm0.00}}$ | $\mathbf{0.18_{\pm0.02}}$ | $\mathbf{0.14_{\pm0.01}}$ | $\mathbf{0.23_{\pm0.01}}$ | $\mathbf{0.15_{\pm0.00}}$ |

Table 1: *Results for UOLS problems under sinusoidal (Sin) and Bernoulli (Ber) shifts.* **Top:** Classification Error. **Bottom:** Mean-squared error in estimating label marginal. For both, lower is better. Across all datasets, we observe that FLH-FTL (ours) often improves over best alternatives.

during offline training and validation, and the unlabeled data that we observe during online learning. We experiment with varying sizes of holdout offline data which is used to obtain the confusion matrix and update the model parameters to adapt to OLS to probe the sample efficiency of all the methods. In contrast to previous works [8, 76], we have chosen to use a smaller amount of holdout offline data for our main experiments. We made this decision because the standard practice for deployment involves training and validating models on training and holdout splits, respectively (e.g., with k-fold cross-validation). Then, the final model is deployed by training on all available data (i.e., the union of train and holdout) with the identified hyperparameters. However, to employ UOLS techniques in practice, practitioners must hold out data that was not seen during training to update the model during online adaptation. Therefore, methods that are efficient with respect to the amount of offline holdout data required might be preferable.

| | Base | Oracle | ROGD | FTH | FTFWH | FLH-FTL (ours) |
|---|---|---|---|---|---|---|
| Cl Err | $18_{\pm1}$ | $6.3_{\pm1.3}$ | $19_{\pm3}$ | $14_{\pm2}$ | $14_{\pm2}$ | $\mathbf{13_{\pm2}}$ |
| MSE | NA | $0.0_{\pm0.0}$ | $0.3_{\pm0.0}$ | $0.3_{\pm0.0}$ | $0.3_{\pm0.0}$ | $\mathbf{0.2_{\pm0.0}}$ |

Table 2: *Results with a Random Forest classifier on MNIST dataset.* Note that methods that update model parameters are not applicable here. FLH-FTL outperforms existing alternatives for both accuracy and label marginal estimation.

| | CT (base) | CT-RS (ours) w FTH | CT-RS (ours) w FLH-FTL | w-ERM (oracle) |
|---|---|---|---|---|
| Cl Err | $20.0_{\pm0.5}$ | $18.38_{\pm0.4}$ | $\mathbf{17.12_{\pm0.8}}$ | $16.32_{\pm0.7}$ |
| MSE | NA | $0.18_{\pm0.01}$ | $\mathbf{0.12_{\pm0.01}}$ | NA |

Table 3: *Results on SOLS setup* on CIFAR10 SOLS with Bernoulli shift. CT with RS improves over the base model (CT) and achieves competitive performance with respect to weighted ERM oracle. MNIST results are similar (see App. F).

For all datasets except SHL, we simulate online label shifts with four types of shifts studied in Bai et al. [8]: monotone shift, square shift, sinusoidal shift, and Bernoulli shift. For SHL locomotion, we use the real-world shift occurring over time. For architectures, we use an MLP for Fashion, SHL and MNIST, Resnets [39] for EuroSAT, CINIC, and CIFAR, and DistilBERT [62, 75] based models for arXiv. For alternate approaches, along with a base classifier (which does no adaptation) and oracle classifier (which reweight using the true label marginals), we make comparisons with adaptation algorithms proposed in prior works [76, 8]. In particular, we compare with ROGD, FTH, FTFWH from Wu et al. [76] and UOGD, ATLAS from Bai et al. [8]. For brevity, we refer to our method as FLH-FTL (though strictly speaking, our methods are based on FLH from Hazan and Seshadri [38] with online averages as base learners). We run all the online label shift experiments with the time horizon $T = 1000$ and at each step 10 samples are revealed. We repeat all experiments with 3 seeds to obtain means and standard deviations of the results. For other methods that perform re-weighting correction on softmax predictions, we use the labeled holdout data to calibrate the model with temperature scaling, which tunes one temperature parameter [35]. We provide exact details about the datasets, label shift simulations, models, and prior methods in App. F.

**Results** Overall, across all datasets, we observe that our method FLH-FTL performs better than alternative approaches in terms of both classification error and mean squared error for estimating the label marginal. Note that methods that directly update the model parameters (i.e., UOGD, ATLAS) do not provide any estimate of the label marginal (Table 1). UOGD and ATLAS also require offline holdout labeled data (i.e., from time step 0) to make online updates to the model parameters. For this purpose, we use the same labeled data that we use to compute the confusion matrix.

As we increase the holdout offline labeled dataset size for updating the model parameters (and to compute the confusion matrix), we observe that classification error and MSE with FLH-FTL stay (relatively) constant whereas the classification errors of other alternatives improve (Fig. 2). This highlights that FLH-FTL can be much more sample efficient with respect to the size of the hold-out offline labeled data. Motivated by this observation, we perform an additional experiment in which we increase the offline training data and observe that we can overall improve the classification accuracy significantly with FLH-FTL (Fig. 2). We present results on SHL dataset with similar findings on semi-synthetic datasets in App. G.5. Finally, we also experiment with a random forest model on the MNIST dataset. Note methods that update model parameters (e.g., UOGD and ATLAS) with OGD are not applicable here. Here, we also observe that we improve over existing applicable alternatives (Table 2).

## 5.2 SOLS setup and results

**Setup** For the supervised problem, we experiment with MNIST and CIFAR datasets. We simulate a time horizon of $T = 200$. For each dataset, at each step, we observe 50 samples with Bernoulli shift. Motivated by our theoretical results with weighted ERM, we propose a simple baseline which continually trains the model at every step instead of starting ERM from scratch every time. We maintain a pool of all the labeled data received till that time step, and at every step, we randomly sample a batch with uniform label marginal to update the model. Finally, we re-weight the updated softmax outputs with estimated label marginal. We call this method Continual Training via Re-Sampling (CT-RS). Its relation as a close variant of weighted ERM is elaborated in App. F.1. To estimate the label marginal, we try FTH and ours FLH-FTL.

**Results** On both datasets, we observe that empirical performance with CT-RS improves over the naive continual training baseline. Additionally, CT-RS results are competitive with weighted ERM

while being 5–15× faster in terms of computation cost (we include the exact computational cost in App. F.1). Moreover, as in UOLS setup, we observe that FLH-FTL improves over FTH for both target label marginal estimation and classification.

## 6 Conclusion

In this work, we focused on unsupervised and supervised online label shift settings. For both settings, we developed algorithms with minimax optimal dynamic regret. Experimental results on both real and semi-synthetic datasets substantiate that our methods improve over prior works both in terms of accuracy and target label marginal estimation.

In future work, we aim to expand our experiments to more real-world label shift datasets. Our work also motivates future work in exploiting other causal structures (e.g. covariate shift) for online distribution shift problems.

## Acknowledgements

SG acknowledges Amazon Graduate Fellowship and JP Morgan AI Ph.D. Fellowship for their support. ZL acknowledges Amazon AI, Salesforce Research, Facebook, UPMC, Abridge, the PwC Center, the Block Center, the Center for Machine Learning and Health, and the CMU Software Engineering Institute (SEI) via Department of Defense contract FA8702-15-D-0002, for their generous support of ACMI Lab's research on machine learning under distribution shift. DB and YW were partially supported by NSF Award #2007117 and a gift from Google.

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

# Appendix

## A   Limitations

Our work is based on the label shift assumption which restricts the applicability of our methods to scenarios where this assumption holds. However, as noted in Section 1, the problem of adaptation to changing data distribution is intractable without imposing assumptions on the nature of the shift.

Furthermore, as noted in Remark 9, our methods in the SOLS settings have a memory requirement that scales linearly with time, which may not be feasible in scenarios where memory is limited. This is reminiscent to FTL / FTRL type algorithms from online learning. We leave the task of deriving theoretical guarantees with reduced storage complexity under non-convex losses as an important future direction.

## B   Related work

**Offline total variation denoising**   The offline problem of Total Variation (TV) denoising constitutes estimating the ground truth under the observation model in Definition 1 with the caveat that all observations are revealed ahead of time. This problem is well studied in the literature of locally adaptive non-parametric regression [20, 21, 23, 52, 69, 45, 66, 73, 60, 34]. The optimal total squared error (TSE) rate for estimation is known to be $\tilde{O}(T^{1/3}V_T^{2/3} + 1)$ [22]. Estimating sequences of bounded TV has received a lot of attention in literature mainly because of the fact that these sequences exhibit spatially varying degree of smoothness. Most signals of scientific interest are known to contain spatially localised patterns [44]. This property also makes the task of designing optimal estimators particularly challenging because the estimator has to efficiently detect localised patterns in the ground truth signal and adjust the amount of smoothing to be applied to optimally trade-off bias and variance.

**Non-stationary online learning**   The problem of online regression can be casted into the dynamic regret minimisation framework of online learning. We assume the notations in Definition 1. In this framework, at each round the learner makes a decision $\hat{\theta}_t$. Then the learner suffers a squared error loss $\ell_t(\hat{\theta}_t) = \|z_t - \hat{\theta}_t\|_2^2$. The gradient of the loss at the point of decision, $\nabla \ell_t(\hat{\theta}_t) = 2(\hat{\theta}_t - z_t)$, is revealed to the learner. The expected dynamic regret against the sequence of comparators $\theta_{1:T}$ is given by

$$
\begin{aligned}
R(\theta_{1:T}) &= \sum_{t=1}^{T} E[\ell_t(\hat{\theta}_t) - \ell_t(\theta_t)] \\
&= \sum_{t=1}^{T} E[\|z_t - \hat{\theta}_t\|_2^2] - E[\|z_t - \theta_t\|_2^2] \\
&= \sum_{t=1}^{T} E\left[\|\hat{\theta}_t\|_2^2 - \|\theta_t\|_2^2 - 2z_t^T\hat{\theta}_t + 2z_t^T\theta_t\right] \\
&= \sum_{t=1}^{T} E[\|\hat{\theta}_t - \theta_t\|_2^2],
\end{aligned}
$$

where in the last line we used the fact that the noise $\epsilon_t$ (see Definition 1) is zero mean and independent of the online decisions $\hat{\theta}_t$. Due to this relation, we conclude that any algorithm that can optimally control the dynamic regret with respect to squared error losses $\ell_t(x) = \|z_t - x\|_2^2$ can be directly used to control the TSE from the ground truth sequence $\theta_{1:T}$.

The minimax estimation rate is defined as follows:

$$
R^*(T, V_T) = \min_{\hat{\theta}_{1:T}} \max_{\substack{\theta_{1:T} \\ \sum_{i=2}^{T}\|\theta_i - \theta_{i-1}\|_1 \leq V_T}} \sum_{t=1}^{T} E[\|\hat{\theta}_t - \theta_t\|_2^2]
$$

Algorithms that can control the dynamic regret with respect to convex losses such as those presented in the works of Jadbabaie et al. [43], Besbes et al. [10], Yang et al. [78], Chen et al. [14], Zhang et al. [81], Goel and Wierman [32], Zhao et al. [84], Zhao and Zhang [83], Zhao et al. [85], Chang

| Algorithm | Run-time | Memory |
|-----------|----------|--------|
| FLH-FTL [38] | $O(T^2)$ | $O(T^2)$ |
| Aligator [7] | $O(T \log T)$ | $O(T)$ |
| Arrows [3] | $O(T \log T)$ | $O(1)$ |

Table 4: Run-time and memory complexity of various adaptively minimax optimal online regression algorithms (see Definition 1). For practical purposes, the storage requirement is negligible even for FLH-FTL. For example, with 10 classes and $T = 1000$, the storage requirement of FLH-FTL is only 40KB, which is insignificant compared to the storage capacity of most modern devices.

and Shahrampour [13], Jacobsen and Cutkosky [42], Baby and Wang [6] can lead to sub-optimal estimation rates of order $O(\sqrt{T(1 + V_T)})$.

On the other hand, algorithms presented in Hazan and Seshadhri [38], Daniely et al. [19], Baby and Wang [3], Baby et al. [7], Raj et al. [56], Baby and Wang [4, 5] exploit the curvature of the losses and attain the (near) optimal estimation rate of $\tilde{O}(T^{1/3}V_T^{2/3} + 1)$.

**Online non-parametric regression**  The task of estimating a sequence of TV bounded sequence from noisy observations can be cast into the online non-parametric regression framework of Rakhlin and Sridharan [57]. Results on online non-parametric regression against reference class of Lipschitz sequences, Sobolev sequences and isotonic sequences can be found in [25, 47, 48] respectively. However as noted in Baby and Wang [3], these classes feature sequences that are more regular than TV bounded sequences. In fact they can be embedded inside a TV bounded sequence class [59] . So the minimax optimality of an algorithm for TV class implies minimax optimality for the smoother sequence classes as well.

**Offline label shift**  Dataset shifts are predominantly studied under two scenarios: covariate shift and label shift [65]. Under covariate shift, $p(y|x)$ doesn't change, while in label shift $p(x|y)$ remains invariant. Schölkopf et al. [63] articulates connections between label shift and covariate shift with anti-causal and causal models respectively. Covariate shift is well explored in past [80, 79, 17, 16, 33]. The offline label shift assumption has been extensively studied in the domain adaptation literature [61, 65, 80, 51, 36, 26, 1, 30] and is also related to problems of estimating mixture proportions of different classes in unlabeled data where previously unseen classes may appear (e.g., positive and unlabeled learning; Elkan and Noto [24], Bekker and Davis [9], Garg et al. [27, 28], Roberts et al. [58]). Classical approaches suffer from the curse of dimensionality, failing in settings with high dimensional data where deep learning prevails. More recent methods work leverage black-box predictors to produce sufficient low-dimensional representations for identifying target distributions of interest [51, 2, 61, 1]. In our work, we leverage black box predictors to estimate label marginals at each step which we track with online regression oracles to trade off variance with (small) bias.

## C  Omitted proofs from Section 3

In the next two lemmas, we verify Assumption 2 for some important loss functions.

**Lemma 11** (cross-entropy loss). *Consider a sample $(x, y) \sim Q$. Let $p \in \mathbb{R}_+^K$ and $\tilde{p}(x) \in \Delta_K$ be a distribution that assigns a weight proportional $\frac{p(i)}{q_0(i)} f_0(i|x)$ to the label $i$ . Let $\ell(\tilde{p}(x), y) = \sum_{i=1}^K \mathbb{I}\{y = i\} \log(1/p(x)[i])$ be the cross-entropy loss. Let $L(p) := E_{(x,y) \sim Q}[\ell(p(x), y)]$ be its population analogue. Then $L(p)$ is $2\sqrt{K}/\mu$ Lipschitz in $\|\cdot\|_2$ norm over the clipped box $\mathcal{D} := \{p \in \mathbb{R}_+^K : \mu \leq p(i) \leq 1 \, \forall i \in [K]\}$ which is compact and convex. Further, the true marginals $q_t \in \mathcal{D}$ whenever $q_t(i) \geq \mu$ for all $i \in [K]$.*

*Proof.* We have

$$L(p) = -\sum_{i=1}^K E[E[Q_t(i|x) \log(\tilde{p}(x)[i])|x]]$$

$$= E[\log(\sum_{i=1}^K w_i p(i))] - \sum_{i=1}^K E[E[Q_t(i|x) \log(w_i p(i))|x]],$$

where we define $w_i := f_0(i|x)/q_0(i)$. Then we can see that

$$\nabla L(p)[i] = E\left[\frac{w_i}{\sum_{j=1}^{K} w_j p(j))}\right] - E\left[\frac{Q_t(i|x)}{p(i)}\right].$$

So if $\min_i p(i) \geq \mu$, we have that $\frac{w_i}{\sum_{j=1}^{K} w_j p(j)} \leq 1/\mu$ and $Q_t(i|x)/p(i) \leq 1/\mu$. So by triangle inequality, $|\nabla L(p)[i]| \leq 1/\mu + 1/\mu$.

$\square$

**Lemma 12** (binary 0-1 loss). *Consider a sample $(x, y) \sim Q$. Let $p \in \mathbb{R}_+^K$ and $\tilde{p}(x) \in \Delta_K$ be a distribution that assigns a weight proportional $\frac{p(i)}{q_0(i)} f_0(i|x)$ to the label $i$. Let $\hat{y}(x)$ be a sample obtained from the distribution $\tilde{p}(x)$. Consider the binary 0-1 loss $\ell(\hat{y}(x), y) = \mathbb{I}(\hat{y}(x) \neq y)$. Let $L(p) := E_{(x,y) \sim Q, \hat{y}(x) \sim \tilde{p}(x)} I(\hat{y}(x) \neq y)$ be its population analogue. Let $q_0(i) \geq \alpha > 0$. Then $L(p)$ is $2K^{3/2}/(\alpha\tau)$ Lipschitz in $\|\cdot\|_2$ norm over the domain $\mathcal{D} := \{p \in \mathbb{R}_+^K : \sum_{i=1}^{K} p(i)f_0(i|x) \geq \tau, p(i) \leq 1 \, \forall i \in [K]\}$ which is compact and convex. Further, the true marginals $q_t \in \mathcal{D}$ whenever $q_t(i) \geq \mu$ for all $i \in [K]$.*

*Proof.* We have that

$$L(p) = \sum_{i=1}^{K} E[Q(y \neq i|x)\tilde{p}(x)[i]].$$

Denote $\tilde{p}(x)[i] = p(i)w_i/\sum_{j=1}^{K} p(j)w_j$ with $w_j := f_0(i|x)/q_0(i)$. Then we see that

$$\left|\frac{\partial \tilde{p}(x)[i]}{\partial p(i)}\right| = \left|\frac{w_i}{\sum_{j=1}^{K} w_j p(j)} - \frac{(w_i p(i))w_i}{\left(\sum_{j=1}^{K} w_j p(j)\right)^2}\right|$$

$$\leq \frac{1}{\alpha\tau} + \frac{w_i}{\sum_{j=1}^{K} w_j p(j)}$$

$$\leq 2/(\alpha\tau).$$

Similarly,

$$\left|\frac{\partial \tilde{p}(x)[i]}{\partial p(j)}\right| = \frac{w_i p(i) w_j}{\left(\sum_{j=1}^{K} w_j p(j)\right)^2}$$

$$\leq 1/(\alpha\tau).$$

Thus we conclude that $\|\nabla \tilde{p}(x)[i]\|_2 \leq 2\sqrt{K}/(\alpha\tau)$, where the gradient is taken with respect to $p \in \mathbb{R}_+^K$.

Therefore,

$$\|\nabla L(p)\|_2 \leq \sum_{i=1}^{K} \|\nabla \tilde{p}(x)[i]\|_2$$

$$\leq 2K^{3/2}/(\alpha\tau).$$

$\square$

**Remark 13.** *The condition $\sum_{i=1}^{K} f_0(i|x)p(i) \geq \tau$ is closely related to Condition 1 of Garg et al. [26]. Note that this is strictly weaker than imposing the restriction that the distribution $p(i) \geq \mu$ for each $i$.*

**Remark 14.** *We emphasize that the conditions in Lemmas 11 and 12 are only sufficient conditions that imply bounded gradients. However, they are not necessary for satisfying bounded gradients property.*

**Lemma 15.** *Let $\mu, \nu \in \Delta_K$ be such that $\mu[i] = q_t(i)$. Let $s_t = C^{-1}f_0(x_t)$, where $C$ is the confusion matrix defined in Assumption 1. We have that $E[s_t] = \mu$ and $Var(s_t) \leq 1/\sigma^2_{min}(C)$*

*Proof.* Let $\tilde{q}_t(\hat{y}_t) = E_{x_t \sim Q_t^X, \hat{y}(x_t) \sim f_0(x_t)} \mathbb{I}\{\hat{y}(x_t) = \hat{y}_t\}$ be the probability that the classifier $f_0$ predicts the label $\hat{y}_t$. Here $Q_t^X(x) := \sum_{i=1}^K Q_t(x, i)$. Let's denote $Q_t(\hat{y}(x_t) = \hat{y}_t | y_t = i) := E_{x_t \sim Q_t(\cdot | y=i), \hat{y}(x_t) \sim f_0(x_t)} \mathbb{I}\{\hat{y}(x_t) = \hat{y}_t\}$. By law of total probability, we have that

$$\tilde{q}_t(\hat{y}_t) = \sum_{i=1}^K Q_t(\hat{y}(x_t) = \hat{y}_t | y_t = i) q_t(i)$$

$$= \sum_{i=1}^K Q_0(\hat{y}(x_t) = \hat{y}_t | y_t = i) q_t(i),$$

where the last line follows by the label shift assumption.

Let $\mu, \nu \in \mathbb{R}^K$ be such that $\mu[i] = q_t(i)$ and $\nu[i] = \tilde{q}_t(i)$. Then the above equation can be represented as $\nu = C\mu$. Thus $\mu = C^{-1}\nu$.

Given a sample $x_t \in Q_t$, the vector $f_0(x_t)$ forms an unbiased estimate of $\nu$. Hence we have that the vector $\hat{\mu} := C^{-1}f_0(x_t)$ is an unbiased estimate of $\mu$. Moreover,

$$\|\hat{\mu}\|_2 \leq \|C^{-1}\|_2 \|f_0(x_t)\|$$
$$\leq 1/\sigma_{min}(C).$$

Hence the variance of the estimate $\hat{\mu}$ is bounded by $1/\sigma^2_{min}(C)$.

$\square$

We have the following performance guarantee for online regression due to Baby et al. [7].

**Proposition 16** (Baby et al. [7])**.** *Let $s_t = C^{-1}f_0(x_t)$. Let $\hat{q}_t := ALG(s_{1:t-1})$ be the online estimate of the true label marginal $q_t$ produced by the Aligator algorithm by taking $s_{1:t-1}$ as input at a round $t$. Then we have that*

$$\sum_{t=1}^T E\left[\|\hat{q}_t - q_t\|_2^2\right] = \tilde{O}(K^{1/3}T^{1/3}V_T^{2/3}(1/\sigma^{4/3}_{min}(C)) + K),$$

*where $V_T := \sum_{t=2}^T \|q_t - q_{t-1}\|_1$. Here $\tilde{O}$ hides dependencies in absolute constants and poly-logarithmic factors of the horizon. Further this result is attained without prior knowledge of the variation $V_T$.*

By following the arguments in Baby and Wang [4], a similar statement can be derived also for the FLH-FTL algorithm of Hazan and Seshadhri [38] (Algorithm 7).

**Theorem 2.** *Suppose we run Algorithm 2 with the online regression oracle ALG as FLH-FTL (App. F) or Aligator [7]. Then under Assumptions 1 and 2, we have*

$$E[R_{dynamic}(T)] = \tilde{O}\left(\frac{K^{1/6}T^{2/3}V_T^{1/3}}{\sigma^{2/3}_{min}(C)} + \frac{\sqrt{KT}}{\sigma_{min}(C)}\right),$$

*where $V_T := \sum_{t=2}^T \|q_t - q_{t-1}\|_1$ and the expectation is taken with respect to randomness in the revealed co-variates. Further, this result is attained without prior knowledge of $V_T$.*

*Proof.* Owing to our carefully crafted reduction from the problem of online label shift to online regression, the proof can be conducted in just a few lines. Let $\tilde{q}_t$ be the value of $ALG(s_{1:t-1})$ computed at line 2 of Algorithm 2. Recall that the dynamic regret was defined as:

$$R_{\text{dynamic}}(T) = \sum_{t=1}^T L_t(\hat{q}_t) - L_t(q_t) \leq \sum_{t=1}^T G\|\hat{q}_t - q_t\|_2 \tag{4}$$

Continuing from Eq.(4), we have

$$
\begin{aligned}
E[R_{\text{dynamic}}(T)] &\leq \sum_{t=1}^{T} G \cdot E[\|\hat{q}_t - q_t\|_2] \\
&\leq \sum_{t=1}^{T} G \cdot E[\|\tilde{q}_t - q_t\|_2] \\
&\leq \sum_{t=1}^{T} G \sqrt{E\|\tilde{q}_t - q_t\|_2^2} \\
&\leq G \sqrt{T \sum_{t=1}^{T} E[\|\tilde{q}_t - q_t\|_2^2]} \\
&= \tilde{O}\left( K^{1/6} T^{2/3} V_T^{1/3} (1/\sigma_{min}^{2/3}(C)) + \sqrt{KT}/\sigma_{min}(C) \right),
\end{aligned}
$$

where the second line is due to non-expansivity of projection, the third line is due to Jensen's inequality, fourth line by Cauchy-Schwartz and last line by Proposition 16. This finishes the proof.

$\square$

Next, we provide matching lower bounds (modulo log factors) for the regret in the unsupervised label shift setting. We start from an information-theoretic result which will play a central role in our lower bound proofs.

**Proposition 17** (Theorem 2.2 in Tsybakov [68]). *Let $\mathbb{P}$ and $\mathbb{Q}$ be two probability distributions on $\mathcal{H}$, such that $KL(\mathbb{P}||\mathbb{Q}) \leq \beta < \infty$, Then for any $\mathcal{H}$-measurable real function $\phi : \mathcal{H} \to \{0, 1\}$,*

$$
\max\{\mathbb{P}(\phi = 1), \mathbb{Q}(\phi = 0)\} \geq \frac{1}{4}\exp(-\beta).
$$

**Theorem 5.** *Let $V_T \leq 64T$. There exists a loss function, a domain $\mathcal{D}$ (in Assumption 2), and a choice of adversarial strategy for generating the data such that for any algorithm, we have $\sum_{t=1}^{T} E([L_t(\hat{q}_t)] - L_t(q_t)) = \Omega\left(\max\{T^{2/3}V_T^{1/3}, \sqrt{T}\}\right)$ , where $\hat{q}_t \in \mathcal{D}$ is the weight estimated by the algorithm and $q_t \in \mathcal{D}$ is the label marginal at round $t$ chosen by the adversary. Here the expectation is taken with respect to the randomness in the algorithm and the adversary.*

*Proof.* We start with a simple observation about KL divergence. Consider distributions with density $P(x, y) = P_0(x|y)p(y)$ and $Q(x, y) = P_0(x|y)q(y)$ where $(x, y) \in \mathbb{R} \times [K]$. Note that these distributions are consistent with the label shift assumption. We note that

$$
\begin{aligned}
KL(P||Q) &= \sum_{i=1}^{K} \int_{\mathbb{R}} P_0(x|i)p(i) \log\left(\frac{P_0(x|i)p(i)}{P_0(x|i)q(i)}\right) dx \\
&= \sum_{i=1}^{K} \int_{\mathbb{R}} P_0(x|i)p(i) \log\left(\frac{p(i)}{q(i)}\right) dx \\
&= \sum_{i=1}^{K} p(i) \log\left(\frac{p(i)}{q(i)}\right)
\end{aligned}
$$

Thus we see that under the label shift assumption, the KL divergence is equal to the KL divergence between the marginals of the labels.

Next, we define a problem instance and an adversarial strategy. We focus on a binary classification problem where the labels is either 0 or 1. As noted before, the KL divergence only depends on the

marginal distribution of labels. So we fix the density $Q_0(x|y)$ to be any density such that under the uniform label marginals ($q_0(1) = q_0(0) = 1/2$) we can find a classifier with invertible confusion matrix (recall from Fig. 1 that $Q_0$ corresponds to the data distribution of the training data set).

Divide the entire time horizon $T$ is divided into batches of size $\Delta$. So there are $M := T/\Delta$ batches (we assume divisibility). Let $\Theta = \left\{ \frac{1}{2} - \delta, \frac{1}{2} + \delta \right\}$ be a set of success probabilities, where each probability can define a Bernoulli trial. Here $\delta \in (0, 1/4)$ which will be tuned later.

The problem instance is defined as follows:

- For batch $i \in [M]$, adversary selects a probability $\mathring{q}_i \in \Theta$ uniformly at random.

- For any round $t$ that belongs to the $i^{th}$ batch, sample a label $y_t \sim \text{Ber}(q_t)$ and co-variate $x_t \sim Q_0(\cdot|y_t)$. Here $q_t = \mathring{q}_i$. The co-variate $x_t$ is revealed.

- Let $\hat{q}_t$ be any estimate of $q_t$ at round $t$. Define the loss as $L_t(\hat{q}_t) := \mathbb{I}\{q_t \geq 1/2\}(1 - \hat{q}_t) + \mathbb{I}\{q_t < 1/2\}\hat{q}_t$.

We take the domain $\mathcal{D}$ in Assumption 2 as $[1/2 - \delta, 1/2 + \delta]$. It is easy to verify that $L_t(\hat{q}_t)$ is Lipschitz over $\mathcal{D}$. Note that unlike Besbes et al. [10], we do not have an unbiased estimate of the gradient of loss functions.

Let's compute an upperbound on the total variation incurred by the true marginals. We have

$$\sum_{t=2}^{T}|q_t - q_{t-1}| = \sum_{i=2}^{M}|\mathring{q}_i - \mathring{q}_{i-1}|$$
$$\leq 2\delta M$$
$$\leq V_T,$$

where the last line is obtained by choosing $\delta = V_T/(2M) = V_T\Delta/(2T)$.

Since at the beginning of each batch, the sampling probability is chosen uniformly at random, the loss function in the current batch is independent of the history available at the beginning of the batch. So only the data in the current batch alone is informative in minimising the loss function in that batch. Hence it is sufficient to consider algorithms that only use the data within a batch alone to make predictions at rounds that falls within that batch.

Now we proceed to bound the regret incurred within batch 1. The computation is identical for any other batches.

Let $\mathbb{P}$ be the joint probability distribution in which labels $(y_1, \ldots, y_\Delta)$ within batch 1 are sampled with success probability $1/2 - \delta$ (i.e $q_t = 1/2 - \delta$)

$$\mathbb{P}(y_1, \ldots, y_\Delta) = \Pi_{i=1}^{\Delta}(1/2 - \delta)^{y_i}(1/2 + \delta)^{1-y_i}.$$

Define an alternate distribution $\mathbb{Q}$ such that

$$\mathbb{Q}(y_1, \ldots, y_\Delta) = \Pi_{i=1}^{\Delta}(1/2 + \delta)^{y_i}(1/2 - \delta)^{1-y_i}.$$

According to the above distribution the data are independently sampled from Bernoulli trials with success probability $1/2 + \delta$. (i.e $q_t = 1/2 + \delta$)

Moving forward, we will show that by tuning $\Delta$ appropriately, any algorithm won't be able to detect between these two alternate worlds with constant probability resulting in sufficiently large regret.

We first bound the KL distance between these two distributions. Let

$$\mathrm{KL}(1/2 - \delta || 1/2 + \delta) := (1/2 + \delta) \log \left( \frac{1/2 + \delta}{1/2 - \delta} \right) + (1/2 - \delta) \log \left( \frac{1/2 - \delta}{1/2 + \delta} \right)$$

$$\leq_{(a)} (1/2 + \delta) \frac{2\delta}{1/2 + \delta} - (1/2 - \delta) \frac{2\delta}{1/2 + \delta}$$

$$= \frac{16\delta^2}{1 - 4\delta^2}$$

$$\leq_{(b)} \frac{64\delta^2}{3},$$

where in line (a) we used the fact that $\log(1 + x) \leq x$ for $x > -1$ and observed that $-4\delta/(1 + 2\delta) > -1$ as $\delta \in (0, 1/4)$. In line (b) we used $\delta \in (0, 1/4)$.

Since $\mathbb{P}$ and $\mathbb{Q}$ are product of the marginals due to independence we have that

$$\mathrm{KL}(\mathbb{P} || \mathbb{Q}) = \sum_{t=1}^{\Delta} \mathrm{KL}(1/2 - \delta || 1/2 + \delta)$$

$$\leq (64\Delta/3) \cdot \delta^2$$

$$= 16/3$$

$$:= \beta, \tag{5}$$

where we used the choices $\delta = \Delta V_T/(2T)$ and $\Delta = (T/V_T)^{2/3}$.

Suppose at the beginning of batch, we reveal the entire observations within that batch $y_{1:\Delta}$ to the algorithm. Note that doing so can only make the problem easier than the sequential unsupervised setting. Let $\hat{q}_t$ be any measurable function of $y_{1:\Delta}$. Define the function $\phi_t := \mathbb{I}\{\hat{q}_t \geq 1/2\}$. Then by Proposition 17, we have that

$$\max\{\mathbb{P}(\phi_t = 1), \mathbb{Q}(\phi_t = 0)\} \geq \frac{1}{4} \exp(-\beta), \tag{6}$$

where $\beta$ is as defined in Eq.(5).

Notice that if $q_t = 1/2 - \delta$, then $L_t(\hat{q}_t) \geq 1/2$ for any $\hat{q}_t \geq 1/2$. Similarly if $q_t = 1/2 + \delta$, we have that $L_t(\hat{q}_t) \geq 1/2$ for any $\hat{q}_t < 1/2$.

Further note that $L_t(q_t) = 1/2 - \delta$ by construction.

For notational clarity define $L_t^p(x) := x$ and $L_t^q(x) := 1 - x$. We can lower-bound the instantaneous regret as:

$$E[L_t(\hat{q}_t)] - L_t(q_t) =_{(a)} \frac{1}{2}(E_{\mathbb{P}}[L_t^p(\hat{q}_t)] - L_t^p(1/2 - \delta)) + \frac{1}{2}(E_{\mathbb{Q}}[L_t^q(\hat{q}_t)] - L_t^q(1/2 + \delta))$$

$$\geq_{(b)} \frac{1}{2}(E_{\mathbb{P}}[L_t^p(\hat{q}_t) | \hat{q}_t \geq 1/2] - L_t^p(1/2 - \delta)\mathbb{P}(\phi_t = 1)$$

$$+ \frac{1}{2}(E_{\mathbb{Q}}[L_t^q(\hat{q}_t) | \hat{q}_t < 1/2] - L_t^q(1/2 + \delta)\mathbb{Q}(\phi_t = 0)$$

$$\geq_{(c)} \frac{1}{2}\delta\mathbb{P}(\phi_t = 1) + \frac{1}{2}\delta\mathbb{Q}(\phi_t = 0)$$

$$\geq \delta/2 \max\{\mathbb{P}(\phi_t = 1), \mathbb{Q}(\phi_t = 0)\}$$

$$\geq_{(d)} \frac{\delta}{8} \exp(-\beta),$$

where in line (a) we used the fact the success probability for a batch is selected uniformly at random from $\Theta$. In line (b) we used the fact that $L_t^p(\hat{q}_t) - L_t^p(1/2 - \delta) \geq 0$ since $\hat{q}_t \in \mathcal{D} = [1/2 - \delta, 1/2 + \delta]$. Similarly term involving $L_t^q$ is also handled. In line (c) we applied $(E_{\mathbb{P}}[L_t^p(\hat{q}_t) | \hat{q}_t \geq 1/2] - L_t^p(1/2 - \delta)) \geq \delta$ since $E_{\mathbb{P}}[L_t^p(\hat{q}_t) | \hat{q}_t \geq 1/2] \geq 1/2$ and $L_t^p(1/2 - \delta) = 1/2 - \delta$. Similar bounding is done for the term involving $E_{\mathbb{Q}}$ as well. In line (d) we used Eq.(6).

---

**Algorithm 5** LPA: a black-box reduction to produce a low-switching online regression algorithm

---

**input** Online regression oracle ALG, failure probability $\delta$, maximum standard deviation $\sigma$ (see
    Definition 1).
1: Initialize prev $= 0 \in \mathbb{R}^K$, $b = 1$
2: Get estimate $\tilde{\theta}_t$ from ALG($z_{1:t-1}$)
3: Output $\hat{\theta}_t =$ prev
4: Receive an observation $z_t$
    // test to detect non-staionarity
5: **if** $\sum_{j=b+1}^{t} \|\text{prev} - \tilde{\theta}_j\|_2^2 > 5K\sigma^2 \log(2T/\delta)$ **then**
6:     Set $b = t + 1$, prev $= z_t$
7:     Restart ALG
8: **else if** $t - b + 1$ is a power of 2 **then**
9:     Set prev $= \sum_{j=b}^{t} z_j / t - b + 1$
10: **end if**
11: Update ALG with $z_t$

---

Thus we get the total expected regret within batch 1 as

$$\sum_{t=1}^{\Delta} E[L_t(\hat{q}_t)] - L_t(q_t) \geq \frac{\delta\Delta}{8}\exp(-\beta)$$

The total regret within any batch $i \in [M]$ can be lower bounded using exactly the same arguments as
above. Hence summing the total regret across all batches yields

$$\sum_{t=1}^{T} E[L_t(\hat{q}_t)] - L_t(q_t) \geq \frac{T}{\Delta} \cdot \frac{\delta\Delta}{8}\exp(-\beta)$$
$$= \frac{V_T\Delta}{16} \cdot \exp(-\beta)$$
$$= T^{2/3}V_T^{1/3}\exp(-\beta)/16.$$

The $\Omega(\sqrt{T})$ part of the lowerbound follows directly from Theorem 3.2.1 in Hazan [37] by choosing
$\mathcal{D}$ with diameter bounded by $\Omega(1)$.

$\square$

# D   Design of low switching online regression algorithms

Even-though Algorithm 4 has attractive performance guarantees, it requires retraining with weighted
ERM at every round. This is not satisfactory since the retraining can be computationally expensive. In
this section, we aim to design a version of Algorithm 4 with few retraining steps while not sacrificing
the statistical efficiency (up to constants). To better understand why this goal is attainable, consider a
time window $[1, n] \subseteq [T]$ where the true label marginals remain constant or drift very slowly. Due to
the slow drift, one reasonable strategy is to re-train the model (with weighted ERM) using the past
data only at time points within $[1, n]$ that are powers of 2 (i.e via a doubling epoch schedule). For
rounds $t \in [1, n]$ that are not powers of 2, we make predictions with a previous model $h_{\text{prev}}$ computed
at $t_{\text{prev}} := 2^{\lfloor \log_2 t \rfloor}$ which is trained using data seen upto the time $t_{\text{prev}}$. Observe that this constitutes
at least half of the data seen until round $t$. This observation when combined with the slow drift of
label marginals implies that the performance of the model $h_{\text{prev}}$ at round $t$ will be comparable to the
performance of a model obtained by retraining using entire data collected until round $t$.

To formalize this idea, we need an efficient online change-point-detection strategy that can detect
intervals where the TV of the *true* label marginals is low and retrain only (modulo at most $\log T$
times within a low TV window) when there is enough evidence for sufficient change in the TV of
the true marginals. We address this problem via a two-step approach. In the first step, we construct
a generic black-box reduction that takes an online regression oracle as input and converts it into

another algorithm with the property that the number of switches in its predictions is controlled without sacrificing the statistical performance. Recall that the purpose of the online regression oracles is to track the true label marginals. The output of our low-switching online algorithm remains the same as long as the TV of the *true* label marginals (TV computed from the time point of the last switch) is sufficiently small. Then we use this low-switching online regression algorithm to re-train the classifier when a switch is detected.

We next provide the **L**ow switching through **P**hased **A**veraging (LPA) (Algorithm 5), our black-box reduction to produce low switching regression oracles. We remark that this algorithm is applicable to the much broader context of *online regression* or *change point detection* and can be of independent interest.

We now describe the intuition behind Algorithm 5. The purpose of Algorithm 5 is to denoise the observations $z_t$ and track the underlying ground truth $\theta_t$ in a statistically efficient manner while incurring low switching cost. Hence it is applicable to the broader context of online non-parametric regression [3, 56, 7] and offline non-parametric regression [66, 72].

Algorithm 5 operates by adaptively detecting low TV intervals. Within each time window it performs a phased averaging in a doubling epoch schedule. i.e consider a low TV window $[b, n]$. For a round $t \in [b, n]$ let $t_{\text{prev}} := 2^{\lfloor \log_2(t-b+1) \rfloor}$. In round $t$, the algorithm plays the average of the observations $z_{b:t_{\text{prev}}}$. So we see that in any low TV window, the algorithm changes its output only at-most $O(\log T)$ times.

For the above scheme to not sacrifice statistical efficiency, it is important to efficiently detect windows with low TV of the true label marginals. Observe that the quantity `prev` computes the average of at-least half of the observations within a time window that start at time $b$. So when the TV of the ground truth within a time window $[b, t]$ is small, we can expect the average to be a good enough representation of the entire ground truth sequence within that time window. Consider the quantity $R_t := \sum_{j=b+1}^{t} \|\text{prev} - \theta_j\|_2^2$ which is the total squared error (TSE) incurred by the fixed decision `prev` within the current time window. Whenever the TV of the ground truth sequence $\theta_{b:t}$ is large, there will be a large bias introduced by `prev` due to averaging. Hence in such a scenario the TSE will also be large indicating non-stationarity. However, we can't compute $R_t$ due to the unavailability of $\theta_j$. So we approximate $R_t$ by replacing $\theta_j$ with the estimates $\tilde{\theta}_j$ coming from the input online regression algorithm that is not constrained by switching cost restrictions. This is the rationale behind the non-stationarity detection test at Step 5. Whenever a non-staionarity is detected we restart the input online regression algorithm as well as the start position for computing averages (in Step 6).

We have the following guarantee for Algorithm 5.

**Theorem 18.** *Suppose the input black box ALG given to Algorithm 5 is adaptively minimax optimal (see Definition 1). Then the number of times Algorithm 5 switches its decision is at most $\tilde{O}(T^{1/3}V_T^{2/3})$ with probability at least $1 - \delta$. Further, Algorithm 5 satisfies $\sum_{t=1}^{T} \|\hat{\theta}_t - \theta_t\|_2^2 = \tilde{O}(T^{1/3}V_T^{2/3})$ with probability at least $1 - \delta$, where $V_T = \sum_{t=2}^{T} \|\theta_t - \theta_{t-1}\|_1$.*

**Remark 19.** *Since Algorithm 5 is a black-box reduction, there are a number of possible candidates for the input policy ALG that are adaptively minimax. Examples include FLH with online averages as base learners [38] or Aligator algorithm [7].*

Armed with a low switching online regression oracle LPA, one can now tweak Algorithm 4 to have sparse number of retraining steps while not sacrificing the statistical efficiency (up to multiplicative constants). The resulting procedure is described in Algorithm 6 (in App. E) which enjoys similar rates as in Theorem 6 (see Theorem 22).

# E   Omitted proofs from Section 4

First we recall a result from Baby et al. [7].

**Proposition 20** (Theorem 5 of Baby et al. [7])**.** *Consider the online regression protocol defined in Definition 1. Let $\hat{\theta}_t$ be the estimate of the ground truth produced by the Aligator algorithm from Baby et al. [7]. Then with probability at-least $1 - \delta$, the total squared error (TSE) of Aligator satisfies*

$$\sum_{t=1}^{T} \|\theta_t - \hat{\theta}_t\|_2^2 = \tilde{O}(T^{1/3}V_T^{2/3} + 1),$$

where $V_T = \sum_{t=2}^{T} \|\theta_t - \theta_{t-1}\|_1$. This bound is attained without any prior knowledge of the variation $V_T$.

The high probability guarantee also implies that

$$\sum_{t=1}^{T} E[\|\theta_t - \hat{\theta}_t\|_2^2] = \tilde{O}(T^{1/3}V_T^{2/3} + 1),$$

where the expectation is taken with respect to randomness in the observations.

By following the arguments in Baby and Wang [4], a similar statement can be derived also for the FLH-FTL algorithm of Hazan and Seshadri [38] (Algorithm 7).

Next, we verify that the noise condition in Definition 1 is satisfied for the empirical label marginals computed at Step 5 of Algorithm 4.

**Lemma 21.** *Let $s_t$ be as in Step 5 of Algorithm 4. Then it holds that $s_t = q_t + \epsilon_t$ with $\epsilon_t$ being independent across $t$ and $\mathrm{Var}(\epsilon_t) \leq 1/N$.*

*Proof.* Since $s_t$ is simply the empirical label proportions, it holds that $E[s_t] = q_t$. Further $\mathrm{Var}(s_t) \leq 1$ as the indicator function is bounded by $1/N$. This concludes the proof. $\square$

**Theorem 6.** *Suppose the true label marginal satisfies $\min_{t,k} q_t(k) \geq \mu > 0$. Choose the online regression oracle in Algorithm 4 as FLH-FTL (App. F) or Aligator from Baby et al. [7] with its predictions clipped such that $\hat{q}_t[k] \geq \mu$. Then with probability at least $1 - \delta$, Algorithm 4 produces hypotheses with $R_{dynamic}^{\mathcal{H}} = \tilde{O}\left(T^{2/3}V_T^{1/3} + \sqrt{T \log(|\mathcal{H}|/\delta)}\right)$, where $V_T = \sum_{t=2}^{T} \|q_t - q_{t-1}\|_1$. Further, this result is attained without any prior knowledge of the variation budget $V_T$.*

*Proof.* In the proof we first proceed to bound the instantaneous regret at round $t$. Re-write the population loss as:

$$L_t(h) = \frac{1}{N(t-1)} \sum_{i=1}^{t-1} \sum_{j=1}^{N} E\left[\frac{q_t(y_{ij})}{q_i(y_{ij})} \ell(h(x_{ij}), y_{ij})\right],$$

where the expectation is taken with respect to randomness in the samples.

We define the following quantities:

$$L_t^{\mathrm{emp}}(h) := \frac{1}{N(t-1)} \sum_{i=1}^{t-1} \sum_{j=1}^{N} \frac{q_t(y_{ij})}{q_i(y_{ij})} \ell(h(x_{ij}), y_{ij}), \tag{7}$$

$$\tilde{L}_t(h) := \frac{1}{N(t-1)} \sum_{i=1}^{t-1} \sum_{j=1}^{N} E\left[\frac{\hat{q}_t(y_{ij})}{\hat{q}_i(y_{ij})} \ell(h(x_{ij}), y_{ij})\right], \tag{8}$$

and

$$\tilde{L}_t^{\mathrm{emp}}(h) := \frac{1}{N(t-1)} \sum_{i=1}^{t-1} \sum_{j=1}^{N} \frac{\hat{q}_t(y_{ij})}{\hat{q}_i(y_{ij})} \ell(h(x_{ij}), y_{ij}).$$

We decompose the regret at round $t$ as

$$L_t(h_t) - L_t(h_t^*) = L_t(h_t) - \tilde{L}_t(h_t) + \tilde{L}_t(h_t) - \tilde{L}_t^{\mathrm{emp}}(h_t) + L_t^{\mathrm{emp}}(h_t^*) - L_t(h_t^*) + \tilde{L}_t^{\mathrm{emp}}(h_t) - L_t^{\mathrm{emp}}(h_t^*)$$

$$\leq \underbrace{L_t(h_t) - \tilde{L}_t(h_t)}_{T1} + \underbrace{\tilde{L}_t(h_t) - \tilde{L}_t^{\mathrm{emp}}(h_t)}_{T2} + \underbrace{L_t^{\mathrm{emp}}(h_t^*) - L_t(h_t^*)}_{T3} + \underbrace{\tilde{L}_t^{\mathrm{emp}}(h_t^*) - L_t^{\mathrm{emp}}(h_t^*)}_{T4},$$

where in the last line we used Eq.(2). Now we proceed to bound each terms as note above.

Note that for any label $m$,

$$
\left| \frac{q_t(m)}{q_i(m)} - \frac{\hat{q}_t(m)}{\hat{q}_i(m)} \right| \leq \left| \frac{q_t(m)}{q_i(m)} - \frac{q_t(m)}{\hat{q}_i(m)} \right| + \left| \frac{q_t(m)}{\hat{q}_i(m)} - \frac{\hat{q}_t(m)}{\hat{q}_i(m)} \right|
$$
$$
\leq \frac{1}{\mu^2} \left( |q_i(m) - \hat{q}_i(m)| + |q_t(m) - \hat{q}_t(m)| \right), \tag{9}
$$

where in the last line, we used the assumption that the minimum label marginals (and hence of the online estimates via clipping) is bounded from below by $\mu$. So by applying triangle inequality and using the fact that the losses are bounded by $B$ in magnitude, we get

$$
T1 \leq \frac{B}{N(t-1)\mu^2} \sum_{i=1}^{t-1} \sum_{j=1}^{N} E\left[ \|\hat{q}_i - q_i\|_1 + \|\hat{q}_t - q_t\|_1 \right]
$$
$$
\leq \frac{B\sqrt{K}}{(t-1)\mu^2} \sum_{i=1}^{t-1} E\left[ \|\hat{q}_i - q_i\|_2 + \|\hat{q}_t - q_t\|_2 \right]
$$
$$
\leq_{(a)} \frac{B\sqrt{K}}{\mu^2} \left( E[\|\hat{q}_t - q_t\|_2] + \sqrt{\frac{\sum_{i=1}^{t-1} E[\|q_i - \hat{q}_i\|_2^2]}{t-1}} \right)
$$
$$
\leq_{(b)} \frac{B\sqrt{K}}{\mu^2} \left( E[\|\hat{q}_t - q_t\|_2] + \phi \cdot \frac{V_T^{1/3}}{(t-1)^{1/3}} \right), \tag{10}
$$

where line (a) is a consequence of Jensen's inequality. In line (b) we used the following fact: by Lemma 21 and Proposition 16, the expected cumulative error of the online oracle at any step is bounded by $\phi t^{1/3} V_t^{2/3}$ for some multiplier $\phi$ which can contain poly-logarithmic factors of the horizon (see Proposition 20).

Proceeding in a similar fashion, the term $T4$ can be bounded by Eq.(10).

Next, we proceed to handle T3. Let $h \in \mathcal{H}$ be any fixed hypothesis. Then each summand in Eq.(7) is an independent random variable assuming values in $[0, B/\mu]$ (recall that the losses lie within $[0, B]$). Hence by Hoeffding's inequality we have that

$$
L_t^{\text{emp}}(h) - L_t(h) \leq \frac{B}{\mu} \sqrt{\frac{\log(3T|\mathcal{H}|/\delta)}{N(t-1)}},
$$
$$
\leq \frac{B}{\mu} \sqrt{\frac{\log(3T|\mathcal{H}|/\delta)}{(t-1)}}, \tag{11}
$$

with probability at-least $1 - \delta/(3T|\mathcal{H}|)$. Now taking union bound across all hypotheses in $\mathcal{H}$, we obtain that:

$$
T3 \leq \frac{B}{\mu} \sqrt{\frac{\log(3|\mathcal{H}|/\delta)}{(t-1)}}, \tag{12}
$$

with probability at-least $1 - \delta/(3T)$.

To bound T2, we notice that it is not possible to directly apply Hoeffding's inequality because the summands in Eq.(8) are correlated through the estimates of the online algorithm. So in the following, we propose a trick to decorrelate them. For any hypothesis $h \in \mathcal{H}$, we have that

$$\frac{\hat{q}_t(y_{ij})}{\hat{q}_i(y_{ij})}\ell(h(x_{ij},y_{ij})) - E\left[\frac{\hat{q}_t(y_{ij})}{\hat{q}_i(y_{ij})}\ell(h(x_{ij},y_{ij}))\right]$$

$$= \underbrace{\left(\frac{\hat{q}_t(y_{ij})}{\hat{q}_i(y_{ij})} - \frac{q_t(y_{ij})}{q_i(y_{ij})}\right)\ell(h(x_{ij},y_{ij}))}_{U_{ij}} -$$

$$\underbrace{E\left[\left(\frac{\hat{q}_t(y_{ij})}{\hat{q}_i(y_{ij})} - \frac{q_t(y_{ij})}{q_i(y_{ij})}\right)\ell(h(x_{ij},y_{ij}))\right]}_{V_{ij}} +$$

$$\underbrace{\frac{q_t(y_{ij})}{q_i(y_ij)}\ell(h(x_{ij},y_{ij})) - E\left[\frac{q_t(y_{ij})}{q_i(y_ij)}\ell(h(x_{ij},y_{ij}))\right]}_{W_{ij}}.$$

Now using Eq.(9) and proceeding similar to the bouding steps of Eq.(10), we obtain

$$\frac{1}{N(t-1)}\sum_{i=1}^{t-1}\sum_{j=1}^{N}U_{ij} \leq \frac{B}{N(t-1)\mu^2}\sum_{i=1}^{t-1}\sum_{j=1}^{N}\|\hat{q}_i - q_i\|_1 + \|\hat{q}_t - q_t\|_1$$

$$\leq \frac{B\sqrt{K}}{\mu^2(t-1)}\sum_{i=1}^{t-1}\|\hat{q}_i - q_i\|_2 + \|\hat{q}_t - q_t\|_2$$

$$\leq_{(a)} \frac{B\sqrt{K}}{\mu^2}\left(\|\hat{q}_t - q_t\|_2 + \sqrt{\frac{\sum_{i=1}^{t-1}\|q_i - \hat{q}_i\|_2^2}{t-1}}\right)$$

$$\leq_{(b)} \frac{B\sqrt{K}}{\mu^2}\left(\|\hat{q}_t - q_t\|_2 + \phi \cdot \frac{V_T^{1/3}}{(t-1)^{1/3}}\right),$$

with probability at-least $1 - \delta/3$. In line (a) we used Jensen's inequaity and in the last line we used the fact the the online oracle attains a high probability bound on the total squared error (TSE) (see Proposition 20).

$\frac{1}{N(t-1)}\sum_{i=1}^{t-1}\sum_{j=1}^{N}V_{ij}$ can be bounded using the same expression as above using similar logic.

To bound $\frac{1}{N(t-1)}\sum_{i=1}^{t-1}\sum_{j=1}^{N}W_{ij}$, we note that it is the sum of independent random variables. Hence using the same arguments used to obtain Eq.(11), we have that

$$\frac{1}{N(t-1)}\sum_{i=1}^{t-1}\sum_{j=1}^{N}W_{ij} \leq \frac{B}{\mu}\sqrt{\frac{\log(3T|\mathcal{H}|/\delta)}{(t-1)}},$$

with probability at-least $1 - \delta/(3T|\mathcal{H}|)$. Hence taking a union bound across all hypothesis classes and across the high probability event of low TSE for the online algorithm yields that

$$T2 \leq \frac{2B\sqrt{K}}{\mu^2}\left(\|\hat{q}_t - q_t\|_2 + \phi \cdot \frac{V_T^{1/3}}{(t-1)^{1/3}}\right) + \frac{B}{\mu}\sqrt{\frac{\log(3T|\mathcal{H}|/\delta)}{(t-1)}},$$

with probability at-least $1 - 2\delta/(3T)$.

Combining the bounds developed for T1,T2,T3 and T4 and by taking a union bound across the event that resulted in Eq.(12), we obtain the following bound on instantaneous regret.

$$L_t(h_t) - L_t(h_t^*) \leq \frac{2B\sqrt{K}}{\mu^2} \left( \|\hat{q}_t - q_t\|_2 + E[\|\hat{q}_t - q_t\|_2] + \phi \cdot \frac{V_T^{1/3}}{(t-1)^{1/3}} + \sqrt{\frac{\log(3T|\mathcal{H}|/\delta)}{(t-1)}} \right),$$

$$(13)$$

with probability at-least $1 - \delta/T$.

Note that via Jensen's inequality:

$$\sum_{t=1}^{T} E[\|q_t - \hat{q}_t\|_2] \leq \sqrt{T \sum_{t=1}^{T} E[\|q_t - \hat{q}_t\|_2^2]}$$

$$\leq \phi T^{2/3} V_T^{1/3},$$

where in the last line we used Proposition 20.

Similarly it can be shown that

$$\sum_{t=1}^{T} \|q_t - \hat{q}_t\|_2 \leq \phi T^{2/3} V_T^{1/3},$$

under the event that resulted in Eq.(13).

Observe that

$$\sum_{t=1}^{T} \frac{V_T^{1/3}}{t^{1/3}} \leq 2T^{2/3} V_T^{1/3}.$$

Finally note that

$$\sum_{t=1}^{T} \frac{1}{\sqrt{t}} \leq 2\sqrt{T}.$$

Hence combining the above bounds and adding Eq.(13) across all time steps, followed by a union bound across all rounds, we obtain that

$$\sum_{t=1}^{T} L_t(h_t) - L_t(h_t^*) \leq \frac{4B\sqrt{K}}{\mu^2} \left( 3\phi T^{2/3} V_T^{1/3} + \sqrt{T \log(3T|\mathcal{H}|/\delta)} \right),$$

with probability at-least $1 - \delta$.

$$\square$$

Next, we prove Theorem 18.

**Theorem 18.** *Suppose the input black box ALG given to Algorithm 5 is adaptively minimax optimal (see Definition 1). Then the number of times Algorithm 5 switches its decision is at most $\tilde{O}(T^{1/3} V_T^{2/3})$ with probability at least $1 - \delta$. Further, Algorithm 5 satisfies $\sum_{t=1}^{T} \|\hat{\theta}_t - \theta_t\|_2^2 = \tilde{O}(T^{1/3} V_T^{2/3})$ with probability at least $1 - \delta$, where $V_T = \sum_{t=2}^{T} \|\theta_t - \theta_{t-1}\|_1$.*

*Proof.* First we proceed to bound the number of switches. Observe that between two time points where condition in Line 5 of Algorithm 5 evaluates true, we can have at-most $\log T$ switches due to the doubling epoch schedule in Line 8.

We first bound the number of times, condition in Line 5 is satisfied. Suppose for some some time $t$, we have that $\sum_{j=b+1}^{t} \|\text{prev} - \tilde{\theta}_j\|_2^2 > 4K\sigma^2 \log(T/\delta)$. Suppose throughout the run of the algorithm, this is $i^{th}$ time the previous condition is satisfied. Let $n_i := t - b + 1$ and let $C_i = \text{TV}[b \to t]$

where $\mathrm{TV}[p \to q] = \sum_{t=p+1}^{q} \|\theta_t - \theta_{t-1}\|_1$. Due to the doubling epoch schedule, we have that that $\mathrm{prev} = \frac{1}{\ell} \sum_{j=b}^{\ell} y_j$ and $E[\mathrm{prev}] = \frac{1}{\ell} \sum_{j=b}^{\ell} \theta_j$ for some $n_i \geq \ell \geq (t - b + 1)/2 = n_i/2$.

So we have

$$\sum_{j=b+1}^{t} \|\mathrm{prev} - \tilde{\theta}_j\|_2^2 \leq \sum_{j=b+1}^{t} 2\|\mathrm{prev} - \theta_j\|_2^2 + 2\|\tilde{\theta}_j - \theta_j\|_2^2$$

$$\leq \sum_{j=b+1}^{t} 2\|E[\mathrm{prev}] - \theta_j\|_2^2 + 2\|\mathrm{prev} - E[\mathrm{prev}]\|_2^2 + 2\|\tilde{\theta}_j - \theta_j\|_2^2$$

$$\leq_{(a)} 2(\ell C_i^2 + 2\sigma^2 K \log(2T/\delta)) + 2\phi n_i^{1/3} C_i^{2/3}$$

$$\leq 4\max\{n_i C_i^2, \phi n_i^{1/3} C_i^{2/3}\} + 4\sigma^2 K \log(2T/\delta)), \tag{14}$$

with probability at-least $1 - \delta/(T)$. In line (a) we used the following facts: i) Due to Hoeffding's inequality, $\|\mathrm{prev} - E[\mathrm{prev}]\|_2^2 \leq \sigma^2 K \log(4T/\delta))/\ell \leq 2\sigma^2 K \log(2T/\delta))/n_i$ with probability at-least $1 - \delta/(2T)$; ii) $\|E[\mathrm{prev}] - \theta_j\|_2 = \|\frac{1}{\ell} \sum_{i=b}^{\ell} \theta_i - \theta_j\|_2 \leq \frac{1}{\ell} \sum_{i=b}^{\ell} \|\theta_i - \theta_j\|_2] \leq C_i$; iii) $\|\tilde{\theta}_j - \theta_j\|_2^2 \leq \phi n_i^{1/3} C_i^{2/3}$ with probability at-least $1 - \delta/(2T)$ due to condition in Theorem 18; iv) Union bound over the events in (i) and (iii).

Since the condition in Line 5 is satisfied at round $t$, Eq.(14) will imply that $5K\sigma^2 \log(2T/\delta) \leq 4\max\{n_i C_i^2, \phi n_i^{1/3} C_i^{2/3}\} + 4\sigma^2 K \log(2T/\delta))$. Rearranging the above, we find that

$$C_i \gtrsim K/\sqrt{n_i},$$

where we suppress the dependence on constants and $\log T$.

Let the condition in Line 5 be satisfied $M$ number of times. By union bound, we have that with probability at-least $1 - \delta$

$$V_T \geq \sum_{i=1}^{M} C_i$$

$$\gtrsim \sum_{i=1}^{M} K/\sqrt{n_i}$$

$$\gtrsim_{(a)} KM \frac{1}{\sqrt{(1/M) \sum_{i=1}^{M} n_i}}$$

$$\gtrsim KM^{3/2}/\sqrt{T},$$

where in Line (a) we used Jensen's inequality. Rearranging we get that

$$M = \tilde{O}(T^{1/3} V_T^{2/3} K^{-2/3}), \tag{15}$$

with probability at-least $1 - \delta$.

Now we proceed to bound the total squared error (TSE) incurred by Algorithm 5. Let $\hat{\theta}_j$ be the output of Algorithm 5 at round $j$. Suppose at times $b - 1$ and $c + 1$, the condition in Line (5) is satisfied. Observe that the condition in Line 5 is not satisfied for any times in $[b, c]$. Then we can conclude that within the interval $[b, c]$ we have that $\sum_{j=b}^{c} \|\hat{\theta}_j - \tilde{\theta}_j\|_2^2 \leq 5K\sigma^2 \log(4T/\delta) \log(T)$, since there are only at-most $\log T$ times within $[b, c]$ where condition in Line 9 is satisfied. So we have that

$$\sum_{j=b}^{c} \|\hat{\theta}_j - \theta_j\|_2^2 \leq \sum_{j=b}^{c} \|\hat{\theta}_j - \tilde{\theta}_j\|_2^2 + \|\theta_j - \tilde{\theta}_j\|_2^2$$

$$\leq 5K\sigma^2 \log(2T/\delta) \log(T) + \phi \cdot n_i^{1/3} C_i^{2/3},$$

with probability at-least $1 - \delta/T$. Here $n_i := b - c + 1$ and $C_i := \text{TV}[b \to c]$. Further we have that $\|\hat{\theta}_{c+1} - \theta_{c+1}\|_2^2 \leq 2B^2$ due to the boundedness condition in Definition 1.

Thus overall we have that $\sum_{j=b}^{c+1} = \tilde{O}(K + n_i^{1/3} C_i^{2/3})$, with probability at-least $1 - \delta$ for any interval [b,c+1] such that condition in Line 5 is satisfied at times $b - 1$ and $c + 1$. Thus we have that

$$
\sum_{t=1}^{T} \|\hat{\theta}_j - \theta_j\|_2^2 \precsim \sum_{i=1}^{M} K + n_i^{1/3} C_i^{2/3}
$$

$$
\precsim_{(a)} T^{1/3} V_T^{2/3} K^{1/3} + \sum_{i=1}^{M} n_i^{1/3} C_i^{2/3}
$$

$$
\precsim_{(b)} T^{1/3} V_T^{2/3} K^{1/3} + \left( \sum_{i=1}^{M} n_i \right)^{1/3} \left( \sum_{i=1}^{M} C_i \right)^{2/3}
$$

$$
\precsim T^{1/3} V_T^{2/3} K^{1/3},
$$

with probability at-least $1 - \delta$. In line (a) we used Eq.(15). In line (b) we used Holder's inequality with the dual norm pair $(3, 3/2)$. This concludes the proof.

$\square$

We now present the tweak of Algorithm 4 by instantiating ALG with Algorithm 5 and prove its regret guarantees. The resulting algorithm is described in Algorithm 6.

---
**Algorithm 6** `Lazy-TrainByWeights`: handling label shift with sparse ERM calls
---
**Input**: Instance ALG of Algorithm 5, A hypothesis Class $\mathcal{H}$
 1: At round $t \in [T]$, get estimated label marginal $\hat{q}_t \in \mathbb{R}^K$ from $\text{ALG}(s_{1:t-1})$.
 2: **if** $\hat{q}_t == \hat{q}_{t-1}$ **then**
 3:      $h_t = h_{t-1}$
 4: **else**
 5:      Update the hypothesis by calling a weighted-ERM oracle:

$$
h_t = \underset{h \in \mathcal{H}}{\arg\min} \sum_{i=1}^{t-1} \sum_{j=1}^{N} \frac{\hat{q}_t(y_{i,j})}{\hat{q}_i(y_{i,j})} \ell(h(x_{i,j}), y_{i,j})
$$

 6: **end if**
 7: Get $N$ co-variates $x_{t,1:N}$ and make predictions according to $h_t$
 8: Get labels $y_{t,1:N}$
 9: Compute $s_t[i] = \frac{1}{N} \sum_{j=1}^{N} \mathbb{I}\{y_{t,j} = i\}$ for all $i \in [K]$.
10: Update ALG with the empirical label marginals $s_t$.

---

**Theorem 22.** *Assume the same notations as in Theorem 6. Suppose we run Algorithm 6 (see Appendix E) with ALG instantiated using Algorithm 5 with $\sigma^2 = 1/N$ and predictions clipped as in Theorem 6. Further let the online regression oracle given to Algorithm 5 be chosen as one of the candidates mentioned in Remark 19. Then with probability at-least $1 - \delta$, we have that*

$$
R_{dynamic}^{\mathcal{H}} = \tilde{O}\left( T^{2/3} V_T^{1/3} + \sqrt{T \log(|\mathcal{H}|/\delta)} \right).
$$

*Further, the number of number of calls to ERM oracle (via Step 5) is at-most $\tilde{O}(T^{1/3} V_T^{2/3})$ with probability at-least $1 - \delta$.*

*Sketch.* The proof of this theorem closely follows the steps fused for proving Theorem 6. So we only highlight the changes that need to be incorporated to the proof of Theorem 6.

Replace the use of Proposition 20 in the proof of Theorem 6 with Theorem 18.

For any round $t$, where Step 5 of Algorithm 6 is triggered, we can use the same arguments as in the Proof of Theorem 22 to bound the instantaneous regret by Eq.(13). i.e:

$$L_t(h_t) - L_t(h_t^*) \le \frac{2B\sqrt{K}}{\mu^2} \left( \|\hat{q}_t - q_t\|_2 + E[\|\hat{q}_t - q_t\|_2] + \phi \cdot \frac{V_T^{1/3}}{(t-1)^{1/3}} + \sqrt{\frac{\log(3T|\mathcal{H}|/\delta)}{(t-1)}} \right),$$
(16)

with probability at-least $1 - \delta/T$.

For a round $t$, where Step 5 is not triggered, we proceed as follows:

Let $t'$ be the most recent time step prior to $t$ when Step 5 is executed. Notice that the population loss can be equivalently represented as

$$L_t(h) = \frac{1}{N(t'-1)} \sum_{i=1}^{t'-1} \sum_{j=1}^{N} E\left[ \frac{q_t(y_{ij})}{q_i(y_{ij})} \ell(h(x_{ij}), y_{ij}) \right],$$

where the expectation is taken with respect to randomness in the samples.

We define the following quantities:

$$L_t^{\text{emp}}(h) := \frac{1}{N(t'-1)} \sum_{i=1}^{t'-1} \sum_{j=1}^{N} \frac{q_t(y_{ij})}{q_i(y_{ij})} \ell(h(x_{ij}), y_{ij}),$$

$$\tilde{L}_t(h) := \frac{1}{N(t'-1)} \sum_{i=1}^{t'-1} \sum_{j=1}^{N} E\left[ \frac{\hat{q}_t(y_{ij})}{\hat{q}_i(y_{ij})} \ell(h(x_{ij}), y_{ij}) \right],$$

and

$$\tilde{L}_t^{\text{emp}}(h) := \frac{1}{N(t'-1)} \sum_{i=1}^{t'-1} \sum_{j=1}^{N} \frac{\hat{q}_t(y_{ij})}{\hat{q}_i(y_{ij})} \ell(h(x_{ij}), y_{ij}).$$

We decompose the regret at round $t$ as

$$L_t(h_t) - L_t(h_t^*) = L_t(h_t) - \tilde{L}_t(h_t) + \tilde{L}_t(h_t) - \tilde{L}_t^{\text{emp}}(h_t) + L_t^{\text{emp}}(h_t^*) - L_t(h_t^*) + \tilde{L}_t^{\text{emp}}(h_t) - L_t^{\text{emp}}(h_t^*)$$

$$\le \underbrace{L_t(h_t) - \tilde{L}_t(h_t)}_{\text{T1}} + \underbrace{\tilde{L}_t(h_t) - \tilde{L}_t^{\text{emp}}(h_t)}_{\text{T2}} + \underbrace{L_t^{\text{emp}}(h_t^*) - L_t(h_t^*)}_{\text{T3}} + \underbrace{\tilde{L}_t^{\text{emp}}(h_t^*) - L_t^{\text{emp}}(h_t^*)}_{\text{T4}},$$

where in the last line we used Eq.(2). Now we proceed to bound each terms as note above.

By using the same arguments as in Proof of Theorem 6 and replacing the use of Proposition 20 with Theorem 18, we can bound T1-4. This will result in an instantaneous regret bound at round $t$ (which doesn't trigger step 5) as:

$$L_t(h_t) - L_t(h_t^*) \le \frac{2B\sqrt{K}}{\mu^2} \left( \|\hat{q}_t - q_t\|_2 + E[\|\hat{q}_t - q_t\|_2] + \phi \cdot \frac{V_T^{1/3}}{(t'-1)^{1/3}} + \sqrt{\frac{\log(3T|\mathcal{H}|/\delta)}{(t'-1)}} \right),$$

$$\le \frac{2B\sqrt{K}}{\mu^2} \left( \|\hat{q}_t - q_t\|_2 + E[\|\hat{q}_t - q_t\|_2] + \phi \cdot 4^{1/3} \cdot \frac{V_T^{1/3}}{(t-1)^{1/3}} + \sqrt{\frac{4\log(3T|\mathcal{H}|/\delta)}{(t-1)}} \right),$$
(17)

with probability at-least $1 - \delta/T$. In the last line we used the fact that $t' - 1 \geq (t/2) - 1 \geq (t-1)/4$ for all $t \geq 3$.

Now adding Eq.(16) and (17) across all rounds and proceeding similar to the proof of Theorem 6 (and replacing the use of Proposition 20 with Theorem 18) completes the argument.

$\square$

We next prove the matching (up to factors of $\log T$) lower bound.

**Theorem 8.** *Let $V_T \leq T/8$. There exists a choice of hypothesis class, loss function, and adversarial strategy of generating the data such that $R^{\mathcal{H}}_{dynamic} = \Omega\left(T^{2/3}V_T^{1/3} + \sqrt{T\log(|\mathcal{H}|)}\right)$, where the expectation is taken with respect to randomness in the algorithm and adversary.*

*Proof.* First we fix the hypothesis class and the data generation strategy. In the problem instance we consider, there are no co-variates. The hypothesis class is defined as

$$\mathcal{H} := \{h_p : h_p \text{ predicts a label } y \sim \text{Ber}(p); \ p \in [|\mathcal{H}|]\}.$$

Further we design the hypothesis class such that both $h_0, h_1 \in \mathcal{H}$. Next we fix the data generation strategy:

- Divide the time horizon into batches of length $\Delta$.

- At the beginning of a batch $i$, the adversary picks $\mathring{q}_i$ uniformly at random from $\{1/2 - \delta, 1/2 + \delta\}$.

- For all rounds $t$ that falls within batch $i$, the label $y_t \sim \text{Ber}(q_t)$ is sampled with $q_t := \mathring{q}_i$.

- Learner predicts a label $\hat{y}_t \in \{0, 1\}$ and then the actual label $y_t$ is revealed (hence $N = 1$ in the protocol of Fig.3).

- Learner suffers a loss given by $\ell_t(\hat{y}_t) = \mathbb{I}\{\hat{y}_t \neq y_t\}$.

It is easy to see that the losses are bounded in $[0, 1]$. Now let's examine the two possibilities of generating labels within a batch. Let's upper bound the variation incurred by the label marginals:

$$\sum_{t=2}^{T} |q_t - q_{t-1}| = \sum_{i=2}^{M} |\mathring{q}_i - \mathring{q}_{i-1}|$$
$$\leq 2\delta M$$
$$\leq V_T,$$

where the last line is obtained by choosing $\delta = V_T/(2M) = V_T\Delta/(2T)$.

Since at the beginning of each batch, the sampling probability of true labels is independently renewed, the historical data till the beginning of a batch is immaterial in minimising the loss within the batch. So we can lower bound the regret within each batch separately and add them up. Below, we focus on lower bounding the regret in batch 1 and the computations are similar for any other batch.

Suppose that the probability that an algorithm predict label $y_t = 1$ is $\hat{q}_t$, where $\hat{q}_t$ is a measurable function of the past data $y_{1:t-1}$. Then we have that the population loss $L_t(\hat{q}_t) := E[\ell_t(\hat{y}_t)] = (1 - \hat{q}_t)q_t + \hat{q}_t(1 - q_t)$. Here we abuse the notation $L(q_t) := L(h_{q_t})$. We see that the population loss $L_t(\hat{q}_t)$ are convex and its gradient obeys $\nabla L_t(\hat{q}_t) = 1 - 2q_t = E[1 - 2y_t]$ since by our construction $y_t \sim \text{Ber}(q_t)$. Thus the population losses are convex and its gradients can be estimated in an unbiased manner from the data.

We use the following Proposition due to Besbes et al. [10].

**Proposition 23** (due to Lemma A-1 in Besbes et al. [10]). *Let $\tilde{\mathbb{P}}$ denote the joint probability of the label sequence $y_{1:\Delta}$ within a batch when they are generated using $Ber(1/2 - \delta)$. So*

$$\tilde{\mathbb{P}}(y_1, \ldots, y_\Delta) = \Pi_{i=1}^{\Delta}(1/2 - \delta)^{y_i}(1/2 + \delta)^{1-y_i}.$$

*Similarly define $\tilde{\mathbb{Q}}$ as*

$$\tilde{\mathbb{Q}}(y_1, \ldots, y_\Delta) = \Pi_{i=1}^\Delta (1/2 + \delta)^{y_i} (1/2 - \delta)^{1-y_i}.$$

*According to the above distribution the data are independently sampled from Bernoulli trials with success probability $1/2 + \delta$. Let $\hat{q}_t$ be the decision of the online algorithm qt round $t$ so that the algorithm predicts label 1 with probability $\hat{q}_t$.*

*Let $\mathbb{P}$ denote the joint probability distribution across the decisions $\hat{q}_{1:\Delta}$ of any online algorithm under the sampling model $\tilde{\mathbb{P}}$. Similarly define $\mathbb{Q}$. Note that any online algorithm can make decisions at round $t$ only based on the past observed data $y_{1:t-1}$. Further after making the decision $\hat{q}_t$ at round $t$, an unbiased estimate of the population loss can be constructed due to the fact that $\nabla L_t(\hat{q}_t) = E[1-2y_t]$. Under the availability of unbiased gradient estimates of the losses, it holds that*

$$KL(\mathbb{P}||\mathbb{Q}) \le 4\Delta\delta^2.$$

*By choosing $\delta = V_T/(2M) = V_T\Delta/(2T)$ and $\Delta = (T/V_T)^{2/3}$, we get that $KL(\mathbb{P}||\mathbb{Q}) \le 1$.*

Since $V_T \le T/8$, the above choice implies that $\delta \in (0, 1/4)$.

For notational clarity, define $L^{\mathbb{P}}(q) = (1-q)(1/2 - \delta) + q(1/2 + \delta)$ and $L^{\mathbb{Q}}(q) = (1-q)(1/2 + \delta) + q(1/2 - \delta)$. These corresponds to the population losses according to the sampling models $\mathbb{P}$ and $\mathbb{Q}$ respectively. Observe that $\min_q L^{\mathbb{P}}(q) = \min_q L^{\mathbb{Q}}(q) = 1/2 - \delta$. The minimum of $L^{\mathbb{P}}$ and $L^{\mathbb{Q}}$ are achieved at 0 and 1 respectively. Note that both $h_0, h_1 \in \mathcal{H}$. So there is always a hypothesis in $\mathcal{H}$ that corresponds the minimiser of the loss.

Further whenever $\hat{q} \ge 1/2$ we have that

$$L^{\mathbb{P}}(q) = (1/2 - \delta) + q(2\delta)$$
$$\ge 1/2.$$

Similarly whenever $q < 1/2$ we have $L^{\mathbb{Q}}(q) \ge 1/2$. So we define the selector function as $\phi_t := \mathbb{I}\{\hat{q}_t \ge 1/2\}$. Let $q_t^* \in \{0, 1\}$ be the minimiser of the loss at round $t$. Now we can lower bound the instantaneous regret similar as

$$E[L_t(\hat{q}_t) - L_t(q_t^*)] = \frac{1}{2}(E_{\mathbb{P}}[L_t^{\mathbb{P}}(\hat{q}_t) - L_t^{\mathbb{P}}(0)] + \frac{1}{2}(E_{\mathbb{Q}}[L_t^{\mathbb{Q}}(\hat{q}_t) - L_t^{\mathbb{Q}}(1)]$$

$$\ge \frac{1}{2}(E_{\mathbb{P}}[L_t^{\mathbb{P}}(\hat{q}_t) - L_t^{\mathbb{P}}(0)|\phi_t = 1]\mathbb{P}(\phi_t = 1) + \frac{1}{2}(E_{\mathbb{Q}}[L_t^{\mathbb{Q}}(\hat{q}_t) - L_t^{\mathbb{Q}}(1)|\phi_t = 0]\mathbb{Q}(\phi_t = 0)$$

$$\ge \delta/2 \max\{\mathbb{P}(\phi_t = 1), \mathbb{Q}(\phi_t = 0)\}$$
$$\ge (\delta/8)e^{-1},$$

where the last line is obtained by Propositions 23 and 17.

Thus we get a total lower bound on the instantanoeus regret as

$$\sum_{t=1}^T E[L_t(\hat{q}_t) - L_t(q_t^*)] \ge T\delta/(8e)$$
$$= \Delta V_T/(16e)$$
$$= T^{2/3}V_T^{1/3}/(16e),$$

where the last line is obtained by using our choices of $\delta V_T\Delta/(2T)$ and $\Delta = (T/V_T)^{2/3}$.

The second term of of $\Omega(\sqrt{T \log|\mathcal{H}|})$ can be obtained from the existing results on statistical learning theory without distribution shifts. (see for example Theorem 3.23 in Mohri et al. [53]).

$\square$

# F   More details on experiments

In Algorithm 7, we describe the FLH-FTL algorithm from Hazan and Seshadhri [38] when specialised to squared error losses. When specialized to squared error losses, this algorithm runs FLH with online averages as the base experts.

---

**Algorithm 7** An instance of FLH-FTL from Hazan and Seshadhri [38] with squared error losses

1: Parameter $\alpha$ is defined to be a learning rate
   `// initializations and definitions`
2: For FLH-FTL instantiations within UOLS algorithms (as in Algorithm 2), we set $\alpha \leftarrow \sigma_{min}^2(C)/(8K)$, where $C$ is the confusion matrix as in Assumption 1. For instantiations within SOLS algorithms (as in Algorithm 4) we set $\alpha \leftarrow 1/(8K)$
3: For each round $t \in [T]$, $v_t := (v_t^{(1)}, \ldots, v_t^{(t)})$ is a probability vector in $\mathbb{R}^t$. Initialize $v_1^{(1)} \leftarrow 1$
4: For each $j \in [T]$, define a base learner $E^j$. For each $t > j$, the base expert outputs $E^j(t) := \frac{1}{t-j}\sum_{i=j}^{t-1} z_j$, where $z_j$ to be specified as below. Further $E^j(j) := 0 \in \mathbb{R}^K$
   `// execution steps`
5: In round $t \in [T]$, set $\forall j \leq t, x_t^j \leftarrow E^j(t)$ (the prediction of the $j^{th}$ base learner at time $t$). Play $x_t = \sum_{j=1}^t v_t^{(j)} x_t^{(j)}$.
6: Receive feedback $z_t$, set $\hat{v}_{t+1}^{(t+1)} \leftarrow 0$ and perform update for $1 \leq i \leq t$:

$$\hat{v}_{t+1}^{(i)} \leftarrow \frac{v_t^{(i)} e^{-\alpha \|x_t^{(i)} - z_t\|_2^2}}{\sum_{j=1}^t v_t^{(j)} e^{-\alpha \|x_t^{(j)} - z_t\|_2^2}}$$

7: Addition step - Set $v_{t+1}^{(t+1)}$ to $1/(t+1)$ and for $i \neq t+1$:

$$v_{t+1}^{(i)} \leftarrow (1 - (t+1)^{-1})\hat{v}_{t+1}^{(i)}$$

---

**Rationale behind the learning rate setting at Line 2 of Algorithm 7** The loss that is incurred by Algorithm 7 and any of its base learners at round $t$ is defined to be the squared error loss $\ell_t(x) = \|z_t - x\|_2^2$. Whenever $\|z_t\|_2^2 \leq B^2$ and $\|x\|_2^2 \leq B^2$, the losses $\ell_t(x)$ are $1/(8B^2)$ exp-concave (see for eg. Chapter 3 of [12]). The notion of exp-concavity is crucial for FLH-FTL algorithm since the learning rate is set to be equal to the exp-concavity factor of the loss functions (see Theorem 3.1 in Hazan and Seshadhri [38]).

For the UOLS problem, from Algorithm 2, we have $\|z_t\|_2 = \|C^{-1}f_0(x_t)\|_2 \leq \sqrt{K}/\sigma_{min}(C)$. Since the decisions of the algorithm is a convex combination of the previously seen $z_t$, we conclude that the losses $\ell_t(x)$ are $\sigma_{min}^2(C)/(8K)$ exp-concave.

For the SOLS problem, let $z_t = s_t$ where $s_t$ is as defined in Algorithm 4. We have that $\|z_t\|_2 \leq \sqrt{K}$. Hence arguing in a similar fashion as above, we conclude that the losses $\ell_t(x)$ are $1/(8K)$ exp-concave for the SOLS problem.

This is the motivation behind Line 2 in Algorithm 7, where the learning rates are set according to the problem setting.

**Dataset and model details.**

- Synthetic: For the synthetic data, we generated 72k samples as described in Bai et al. [8]. There are three classes each with 24k samples generated from three Gaussian distributions in $\mathbb{R}^{12}$. Each Gaussian distribution is defined by a randomly generated unit-norm centre $v$ and covariance matrix $0.215 \cdot I$. 60k samples are used as source data, and 12k samples are used as target data to be sampled from during online learning. We used logistic regression to train a linear model. It is trained for a single epoch with learning rate 0.1, momentum 0.9, batch size 200, and $l_2$ regularization $1 \times 10^{-4}$.

- MNIST [50]: An image dataset of 10 types of handwritten digits. 60k samples are used as source data and 10k as target data. We used an MLP for prediction with three consecutive

hidden layers of sizes 100, 100, and 20. It is trained for a single epoch with a learning rate 0.1, momentum 0.9, batch size 200, and $l_2$ regularization $1 \times 10^{-4}$.

- CIFAR-10 [49]: A dataset of colored images of 10 items: airplane, automobile, bird, cat, deer, dog, frog, horse, ship, and truck. 50k samples are used as source data and 10k as target data. We train a ResNet18 model (He et al. [39]) from scratch. It is finetuned for 70 epochs with learning rate 0.1, momentum 0.9, batch size 200, and $l_2$ regularization $1 \times 10^{-4}$. The learning rate decayed by $90\%$ at the 25th and 40th epochs.

- Fashion [77]: An image dataset of 10 types of fashion items: T-shirt, trouser, pullover, dress, coat, sandals, shirt, sneaker, bag, and ankle boots. 60k samples are used as source data and 10k as target data. We trained an MLP for prediction. It is trained for 50 epochs with learning rate 0.1, momentum 0.9, batch size 200, and $l_2$ regularization $1 \times 10^{-4}$.

- EuroSAT [40]: An image dataset of 10 types of land uses: industrial buildings, residential buildings, annual crop, permanent crop, river, sea & lake, herbaceous vegetation, highway, pasture, and forest. 60k samples are used as source data and 10k as target data. We cropped the images to the size $(3, 64, 64)$. We train a ResNet18 model for 50 epochs with learning rate 0.1, momentum 0.9, batch size 200, and $l_2$ regularization $1 \times 10^{-4}$.

- Arxiv [15]: A natural language dataset of 23 classes over different publication subjects. 198k samples are used as source data and 22k as target data. We trained a DistilBERT model (Sanh et al. [62]) for 50 epochs with learning rate $2 \times 10^{-5}$, batch size 64, and $l_2$ regularization $1 \times 10^{-2}$.

- SHL [31, 71]: A tabular locomotion dataset of 6 classes of human motion: still, walking, run, bike, car and bus. 30k samples are used as source data and 70k as target data. We trained an MLP for prediction for 50 epochs with learning rate 0.1, momentum 0.9, batch size 200, and $l_2$ regularization $1 \times 10^{-4}$.

For all the datasets above, the initial offline data are further split by $80 : 20$ into training and holdout data, where the former is used for offline training of the base model and the latter for computing the confusion matrix and retraining (e.g. updating the linear head parameters with UOGD or updating the softmax prediction with our FLT-FTL) during online learning. To examine how well the algorithms adapt when holdout data is limited, we use $10\%$ of the holdout data (i.e., $2\%$ of the initial offline data) in the main paper unless stated otherwise. In App. G.2, we ablate with full hold-out data.

**Types of Simulated Shifts.** We simulate four kinds of label shifts to capture different non-stationary environments. These shifts are similar to the ones used in Bai et al. [8]. For each shift, the label marginals are a mixture of two different constant marginals $\mu_1, \mu_2 \in \Delta_K$ with a time-varying coefficient $\alpha_t$: $\mu_{y_t} = (1 - \alpha_t)\mu_1 + \alpha_t\mu_2$, where $\mu_{y_t}$ denotes the label distribution at round $t$ and $\alpha_t$ controls the shift non-stationarity and patterns. In particular, we have: *Sinusoidal Shift (Sin)*: $\alpha_t = \sin\frac{i\pi}{L}$, where $i = t \mod L$ and $L$ is a given periodic length. In the experiments, we set $L = \sqrt{T}$. *Bernoulli Shift (Ber)*: at every iteration, we keep the $\alpha_t = \alpha_{t-1}$ with probability $p \in [0, 1]$ and otherwise set $\alpha_t = 1 - \alpha_{t-1}$. In the experiments, the parameter is set as $p = 1/\sqrt{T}$, which implies $V_t = \Theta(\sqrt{T})$. *Square Shift (Squ)*: at every $L$ rounds we set $\alpha_t = 1 - \alpha_{t-1}$. *Monotone Shift (Mon)*: we set $\alpha_t = t/T$. Square, sinusoidal, and Bernoulli shifts simulate fast-changing environments with periodic patterns.

**Methods for UOLS Adaptation.**

- Base: the base classifier without any online modifications.

- OFC: the optimal fixed classifier predicts with an optimal fixed re-weighting in hindsight as in Wu et al. [76].

- Oracle: re-weight the base model's predictions with the true label marginal of the unlabeled data at each time step.

- FTH: proposed by Wu et al. [76], follow-the-history classifier re-weights the base model's predictions with a simple online average of all marginal estimates seen thus far.

- FTFWH: proposed by Wu et al. [76], follow-the-fixed-window-history classifier is a version of FTH that tracks only the $k$ most recent marginal estimates. We choose $k = 100$ throughout the experiments in this work.

| | | CT (base) | CT-RS (ours) w FLH | CT-RS (ours) w FLT-FTL | w-ERM (oracle) |
|---|---|---|---|---|---|
| MNIST | Cl Err | $5.0_{\pm 0.5}$ | $4.71_{\pm 0.2}$ | $\mathbf{4.53_{\pm 0.1}}$ | $3.2_{\pm 0.4}$ |
| | MSE | NA | $0.12_{\pm 0.01}$ | $\mathbf{0.08_{\pm 0.01}}$ | NA |

Table 5: *Results on SOLS setup.* We report results with MNIST SOLS setup runs for $T = 200$ steps. We observe that continual training with re-sampling improves over the base model which continually trains on the online data and achieves competitive performance with respect to weighted ERM oracle.

| | CT-RS (ours) | w-ERM (oracle) |
|---|---|---|
| CIFAR | $\mathbf{145_{\pm 3.7}}$ | $1882_{\pm 14}$ |
| MNIST | $\mathbf{20_{\pm 2.7}}$ | $107_{\pm 3.6}$ |

Table 6: *Comparison on computation time (in minutes).* We report results with MNIST and CIFAR SOLS setup runs for $T = 200$ steps. We observe that continual training with re-sampling is approximately 5–15× more efficient than weighted ERM oracle.

- ROGD: proposed by Wu et al. [76], ROGD uses online gradient descent to update its re-weighting vector based on current marginal estimate.
- UOGD: proposed by Bai et al. [8], retrains the last linear layer of the model based on current marginal estimate.
- ATLAS: proposed by Bai et al. [8] is a meta-learning algorithm that has UOGD as its base learners.

The learning rates of ROGD, UOGD, and ATLAS are set according to suggestions in the original works. The learning rate of FLH-FTL is set to $1/K$. This corresponds to a faster rate than the theoretically optimal learning rate given in Line 2 of Algorithm 7. It has been observed in prior works such as Baby et al. [7] that the theoretical learning rate is often too conservative and faster rates lead to a better performance.

### F.1 Supervised Online Label Shift Experiment Details

For each dataset, we first fix the number of time steps and then simulate the label marginal shift. To train the learner with all the methods, we store all the online data observed giving the storage complexity of $\mathcal{O}(T)$. We observe $N = 50$ examples at every iteration and we split the observed labeled examples into 80:20 split for training and validation. The validation examples are used to decide the number of gradient steps at every time step, in particular, we take gradient steps till the validation error continues to decrease.

**Dataset and model details.**

- MNIST [50]: An image dataset of 10 types of handwritten digits. At each step, we sample 50 samples with the label marginal that step without replacement and reveal the examples to the learner. We used an MLP for prediction with three consecutive hidden layers of sizes 100, 100, and 20. It is trained for a single epoch with a learning rate 0.1, momentum 0.9, batch size 200, and $l_2$ regularization $1 \times 10^{-4}$.
- CIFAR-10 [49]: A dataset of colored images of 10 items: airplane, automobile, bird, cat, deer, dog, frog, horse, ship, and truck. At each step, we sample 50 samples with the label marginal that step without replacement and reveal the examples to the learner. It is finetuned for 70 epochs with learning rate 0.1, momentum 0.9, batch size 200, and $l_2$ regularization $1 \times 10^{-4}$.

We simulate Bernoulli label shifts to capture different non-stationary environments.

**Connection of CT-RS to weighted ERM** Before making the connection, we first re-visit the CT-RS algorithm. **Step 1**: Maintain a pool of all the labeled data received till that time step, and at every

iteration, we randomly sample a batch with uniform label marginal to update the model. **Step 2**: Re-weight the softmax outputs of the updated model with estimated label marginal. Below we show that it is equivalent to wERM:

$$
\begin{aligned}
f_t &= \operatorname*{argmin}_{f \in \mathcal{H}} \sum_{i=1}^{t-1} \sum_{j=1}^{N} \frac{\hat{q}_t(y_{i,j})}{\hat{q}_i(y_{i,j})} \ell(f(x_{i,j}), y_{i,j}) \\
&= \operatorname*{argmin}_{f \in \mathcal{H}} \sum_{k=1}^{K} \hat{q}_t(k) \sum_{i=1}^{t-1} \sum_{j=1}^{N} \frac{\mathbb{1}(y_{i,j}=k)}{\hat{q}_i(k)} \ell(f(x_{i,j}), k) \\
&= \operatorname*{argmin}_{f \in \mathcal{H}} \sum_{k=1}^{K} \frac{\hat{q}_t(k)}{(1/K)} \sum_{i=1}^{t-1} \sum_{j=1}^{N} \frac{\mathbb{1}(y_{i,j}=k)}{K \cdot \hat{q}_i(k)} \ell(f(x_{i,j}), k) \\
&= \operatorname*{argmin}_{f \in \mathcal{H}} \sum_{k=1}^{K} \hat{\mu}_{t,k} \underbrace{\sum_{i=1}^{t-1} \sum_{j=1}^{N} \frac{\mathbb{1}(y_{i,j}=k)}{\hat{\mu}_{i,k}} \ell(f(x_{i,j}), k)}_{L_{t-1,k}}
\end{aligned}
$$

where $\hat{\mu}_{t,k} = \hat{q}_t(k)/(1/K)$ is the importance ratio at time $i$ with respect to uniform label marginal. Similarly, we define $\hat{\mu}_{i,k} = \hat{q}_i(k)/(1/K)$. Here, $L_{t-1,k}$ is the aggregate loss at $t$-th time step for $k$-th class such that at each step the sampling probability of that class is uniform. By continually training a classifier with CT-RS, Step 1 approximates the classifier $\widetilde{f}_t$ trained to minimize the average of $L_{t-1,k}$ over all classes with uniform proportion for each class. To update the classifier $\widetilde{f}_t$ according to label proportions at time $t$, we update the softmax output of $\widetilde{f}_t$ according to $\hat{\mu}_t$.

The primary benefit of CT-RS over wERM is to avoid re-training from scratch at every iteration. Instead, we can leverage the model trained in the previous iteration to warm-start training in the next iteration.

# G Additional Unsupervised Online Label Shift Experiments

## G.1 Additional results with Monotone and Square Shifts and Low Amount of Holdout Data

| Methods | Synthetic | | MNIST | | CIFAR | | EuroSAT | | Fashion | | ArXiv | |
|---|---|---|---|---|---|---|---|---|---|---|---|---|
| | Mon | Squ | Mon | Squ | Mon | Squ | Mon | Squ | Mon | Squ | Mon | Squ |
| Base | $8.7_{\pm0.1}$ | $8.5_{\pm0.2}$ | $4.7_{\pm0.0}$ | $4.4_{\pm0.2}$ | $17_{\pm0}$ | $17_{\pm0}$ | $13_{\pm0}$ | $13_{\pm0}$ | $15_{\pm0}$ | $15_{\pm0}$ | $22_{\pm0}$ | $21_{\pm0}$ |
| OFC | $6.9_{\pm0.1}$ | $6.6_{\pm0.3}$ | $4.1_{\pm0.1}$ | $3.9_{\pm0.2}$ | $14_{\pm0}$ | $14_{\pm0}$ | $11_{\pm1}$ | $11_{\pm0}$ | $9.0_{\pm0.0}$ | $9.6_{\pm0.5}$ | $18_{\pm1}$ | $18_{\pm0}$ |
| Oracle | $5.2_{\pm0.2}$ | $3.6_{\pm0.2}$ | $2.5_{\pm0.1}$ | $2.2_{\pm0.1}$ | $7.7_{\pm0.1}$ | $6.8_{\pm0.2}$ | $5.3_{\pm0.2}$ | $4.4_{\pm0.0}$ | $5.1_{\pm0.1}$ | $4.1_{\pm0.1}$ | $6.9_{\pm0.3}$ | $6.6_{\pm0.2}$ |
| FTH | $7.1_{\pm0.3}$ | $6.8_{\pm0.4}$ | $4.1_{\pm0.1}$ | $\mathbf{4.0_{\pm0.0}}$ | $13_{\pm1}$ | $13_{\pm0}$ | $11_{\pm0}$ | $11_{\pm0}$ | $9.3_{\pm0.6}$ | $8.9_{\pm0.4}$ | $19_{\pm1}$ | $18_{\pm0}$ |
| FTFWH | $\mathbf{6.3_{\pm0.2}}$ | $7.0_{\pm0.0}$ | $\mathbf{4.0_{\pm0.0}}$ | $4.1_{\pm0.1}$ | $12_{\pm0}$ | $13_{\pm0}$ | $9.9_{\pm0.2}$ | $11_{\pm0}$ | $\mathbf{8.4_{\pm0.3}}$ | $9.1_{\pm0.5}$ | $\mathbf{18_{\pm1}}$ | $18_{\pm0}$ |
| ROGD | $7.8_{\pm0.3}$ | $7.8_{\pm0.3}$ | $4.5_{\pm0.2}$ | $5.4_{\pm1.7}$ | $14_{\pm1}$ | $15_{\pm0}$ | $11_{\pm0}$ | $14_{\pm1}$ | $8.9_{\pm0.4}$ | $10_{\pm1}$ | $19_{\pm1}$ | $21_{\pm1}$ |
| UOGD | $8.1_{\pm0.3}$ | $8.1_{\pm0.5}$ | $4.9_{\pm0.1}$ | $4.8_{\pm0.4}$ | $15_{\pm1}$ | $15_{\pm0}$ | $10_{\pm1}$ | $11_{\pm1}$ | $11_{\pm2}$ | $12_{\pm2}$ | $20_{\pm1}$ | $19_{\pm0}$ |
| ATLAS | $8.0_{\pm0.0}$ | $8.2_{\pm0.5}$ | $4.6_{\pm0.2}$ | $4.5_{\pm0.3}$ | $15_{\pm1}$ | $15_{\pm0}$ | $10_{\pm1}$ | $11_{\pm1}$ | $12_{\pm2}$ | $12_{\pm1}$ | $20_{\pm1}$ | $19_{\pm1}$ |
| FLH-FTL (ours) | $\mathbf{6.3_{\pm0.3}}$ | $\mathbf{5.6_{\pm0.4}}$ | $\mathbf{4.0_{\pm0.0}}$ | $\mathbf{4.0_{\pm0.0}}$ | $\mathbf{12_{\pm0}}$ | $\mathbf{12_{\pm0}}$ | $10_{\pm0}$ | $\mathbf{10_{\pm0}}$ | $8.6_{\pm0.4}$ | $\mathbf{8.4_{\pm0.4}}$ | $\mathbf{18_{\pm1}}$ | $\mathbf{17_{\pm0}}$ |

| | Synthetic | | MNIST | | CIFAR | | EuroSAT | | Fashion | | ArXiv | |
|---|---|---|---|---|---|---|---|---|---|---|---|---|
| | Mon | Squ | Mon | Squ | Mon | Squ | Mon | Squ | Mon | Squ | Mon | Squ |
| FTH | $0.11_{\pm0.00}$ | $0.21_{\pm0.01}$ | $0.14_{\pm0.00}$ | $0.27_{\pm0.00}$ | $0.15_{\pm0.01}$ | $0.28_{\pm0.00}$ | $0.14_{\pm0.01}$ | $0.28_{\pm0.00}$ | $0.16_{\pm0.02}$ | $0.28_{\pm0.01}$ | $0.18_{\pm0.00}$ | $0.30_{\pm0.00}$ |
| FTFWH | $\mathbf{0.05_{\pm0.00}}$ | $0.23_{\pm0.01}$ | $\mathbf{0.07_{\pm0.00}}$ | $0.30_{\pm0.00}$ | $\mathbf{0.07_{\pm0.00}}$ | $0.30_{\pm0.00}$ | $\mathbf{0.07_{\pm0.00}}$ | $0.30_{\pm0.00}$ | $\mathbf{0.08_{\pm0.01}}$ | $0.31_{\pm0.00}$ | $\mathbf{0.09_{\pm0.00}}$ | $0.32_{\pm0.00}$ |
| ROGD | $0.18_{\pm0.01}$ | $0.29_{\pm0.01}$ | $0.28_{\pm0.05}$ | $0.41_{\pm0.04}$ | $0.22_{\pm0.04}$ | $0.37_{\pm0.03}$ | $0.27_{\pm0.03}$ | $0.41_{\pm0.02}$ | $0.21_{\pm0.01}$ | $0.37_{\pm0.01}$ | $0.21_{\pm0.01}$ | $0.36_{\pm0.01}$ |
| FLH-FTL (ours) | $\mathbf{0.05_{\pm0.00}}$ | $\mathbf{0.11_{\pm0.00}}$ | $\mathbf{0.07_{\pm0.00}}$ | $\mathbf{0.15_{\pm0.00}}$ | $0.09_{\pm0.01}$ | $\mathbf{0.17_{\pm0.00}}$ | $0.08_{\pm0.01}$ | $\mathbf{0.17_{\pm0.01}}$ | $0.09_{\pm0.01}$ | $\mathbf{0.18_{\pm0.02}}$ | $0.11_{\pm0.00}$ | $\mathbf{0.24_{\pm0.00}}$ |

Table 7: *Results for UOLS problems with monotone and square shifts using low amount of holdout data*. **Top:** Classification Error. **Bottom:** Mean-squared error in estimating label marginal.

## G.2 Additional results with All of Holdout Data

| Methods | Synthetic | | MNIST | | CIFAR | | EuroSAT | | Fashion | | ArXiv | |
|---|---|---|---|---|---|---|---|---|---|---|---|---|
| | Ber | Sin | Ber | Sin | Ber | Sin | Ber | Sin | Ber | Sin | Ber | Sin |
| Base | $8.6_{\pm0.3}$ | $8.2_{\pm0.3}$ | $3.8_{\pm0.3}$ | $3.9_{\pm0.0}$ | $17_{\pm0}$ | $16_{\pm0}$ | $13_{\pm0}$ | $13_{\pm0}$ | $15_{\pm0}$ | $15_{\pm0}$ | $23_{\pm0}$ | $19_{\pm0}$ |
| OFC | $6.7_{\pm0.2}$ | $5.5_{\pm0.2}$ | $3.4_{\pm0.4}$ | $3.4_{\pm0.2}$ | $13_{\pm0}$ | $11_{\pm0}$ | $11_{\pm1}$ | $9.8_{\pm1.3}$ | $8.3_{\pm0.5}$ | $6.8_{\pm0.2}$ | $21_{\pm1}$ | $14_{\pm0}$ |
| Oracle | $3.7_{\pm0.1}$ | $3.7_{\pm0.1}$ | $1.7_{\pm0.2}$ | $1.5_{\pm0.1}$ | $6.3_{\pm0.1}$ | $5.9_{\pm0.1}$ | $4.0_{\pm0.0}$ | $4.1_{\pm0.1}$ | $3.5_{\pm0.1}$ | $3.6_{\pm0.1}$ | $7.8_{\pm0.2}$ | $5.1_{\pm0.1}$ |
| FTH | $6.8_{\pm0.2}$ | $5.5_{\pm0.3}$ | $\mathbf{3.2_{\pm0.2}}$ | $\mathbf{3.2_{\pm0.2}}$ | $12_{\pm0}$ | $\mathbf{10_{\pm0}}$ | $11_{\pm0}$ | $9.5_{\pm0.1}$ | $8.0_{\pm0.0}$ | $\mathbf{6.8_{\pm0.2}}$ | $20_{\pm0}$ | $\mathbf{14_{\pm0}}$ |
| FTFWH | $6.6_{\pm0.3}$ | $5.5_{\pm0.2}$ | $3.3_{\pm0.2}$ | $\mathbf{3.2_{\pm0.1}}$ | $12_{\pm0}$ | $\mathbf{10_{\pm0}}$ | $10_{\pm0}$ | $9.4_{\pm0.2}$ | $7.9_{\pm0.0}$ | $6.9_{\pm0.2}$ | $20_{\pm0}$ | $\mathbf{14_{\pm0}}$ |
| ROGD | $7.8_{\pm0.3}$ | $7.2_{\pm0.3}$ | $4.7_{\pm0.3}$ | $3.3_{\pm0.2}$ | $15_{\pm0}$ | $11_{\pm0}$ | $11_{\pm0}$ | $10_{\pm0}$ | $14_{\pm5}$ | $8.2_{\pm0.2}$ | $23_{\pm0}$ | $16_{\pm1}$ |
| UOGD | $7.6_{\pm0.4}$ | $7.0_{\pm0.0}$ | $\mathbf{3.2_{\pm0.2}}$ | $\mathbf{3.2_{\pm0.2}}$ | $\mathbf{11_{\pm0}}$ | $\mathbf{10_{\pm0}}$ | $\mathbf{7.7_{\pm0.0}}$ | $\mathbf{7.3_{\pm0.2}}$ | $9.6_{\pm0.2}$ | $8.6_{\pm0.1}$ | $\mathbf{19_{\pm0}}$ | $\mathbf{14_{\pm0}}$ |
| ATLAS | $7.5_{\pm0.3}$ | $6.8_{\pm0.3}$ | $\mathbf{3.2_{\pm0.3}}$ | $\mathbf{3.2_{\pm0.2}}$ | $12_{\pm0}$ | $11_{\pm0}$ | $9.1_{\pm0.0}$ | $8.3_{\pm0.2}$ | $12_{\pm0}$ | $11_{\pm0}$ | $21_{\pm0}$ | $16_{\pm0}$ |
| FLH-FTL (ours) | $\mathbf{5.4_{\pm0.3}}$ | $\mathbf{5.3_{\pm0.2}}$ | $\mathbf{3.2_{\pm0.2}}$ | $3.3_{\pm0.2}$ | $\mathbf{11_{\pm0}}$ | $\mathbf{10_{\pm0}}$ | $9.4_{\pm0.2}$ | $9.3_{\pm0.1}$ | $\mathbf{7.5_{\pm0.1}}$ | $7.0_{\pm0.0}$ | $\mathbf{19_{\pm0}}$ | $\mathbf{14_{\pm0}}$ |

| | Synthetic | | MNIST | | CIFAR | | EuroSAT | | Fashion | | ArXiv | |
|---|---|---|---|---|---|---|---|---|---|---|---|---|
| | Ber | Sin | Ber | Sin | Ber | Sin | Ber | Sin | Ber | Sin | Ber | Sin |
| FTH | $0.20_{\pm0.00}$ | $0.10_{\pm0.00}$ | $0.25_{\pm0.00}$ | $0.14_{\pm0.00}$ | $0.28_{\pm0.00}$ | $0.14_{\pm0.00}$ | $0.27_{\pm0.00}$ | $0.14_{\pm0.00}$ | $0.27_{\pm0.00}$ | $0.14_{\pm0.00}$ | $0.29_{\pm0.00}$ | $0.15_{\pm0.00}$ |
| FTFWH | $0.19_{\pm0.00}$ | $0.09_{\pm0.00}$ | $0.24_{\pm0.00}$ | $0.13_{\pm0.00}$ | $0.24_{\pm0.00}$ | $\mathbf{0.13_{\pm0.00}}$ | $0.26_{\pm0.00}$ | $\mathbf{0.13_{\pm0.00}}$ | $0.24_{\pm0.00}$ | $\mathbf{0.13_{\pm0.00}}$ | $0.28_{\pm0.00}$ | $\mathbf{0.15_{\pm0.00}}$ |
| ROGD | $0.29_{\pm0.00}$ | $0.23_{\pm0.00}$ | $0.43_{\pm0.00}$ | $0.33_{\pm0.00}$ | $0.31_{\pm0.00}$ | $0.21_{\pm0.00}$ | $0.41_{\pm0.00}$ | $0.34_{\pm0.00}$ | $0.45_{\pm0.08}$ | $0.31_{\pm0.00}$ | $0.34_{\pm0.00}$ | $0.28_{\pm0.00}$ |
| FLH-FTL (ours) | $\mathbf{0.09_{\pm0.00}}$ | $\mathbf{0.08_{\pm0.00}}$ | $\mathbf{0.13_{\pm0.00}}$ | $\mathbf{0.12_{\pm0.00}}$ | $\mathbf{0.15_{\pm0.00}}$ | $\mathbf{0.13_{\pm0.00}}$ | $\mathbf{0.15_{\pm0.00}}$ | $\mathbf{0.13_{\pm0.00}}$ | $\mathbf{0.15_{\pm0.00}}$ | $\mathbf{0.13_{\pm0.00}}$ | $\mathbf{0.22_{\pm0.00}}$ | $\mathbf{0.15_{\pm0.00}}$ |

Table 8: *Results for UOLS problems using all hold-out data*. **Top:** Classification Error. **Bottom:** Mean-squared error in estimating label marginal. Compared to the result in the main paper (Table 1), we observe that the performances of ROGD, UOGD, and ATLAS depend more on availability of holdout data that FLH-FTL. Notably, UOGD becomes competitive in the majority of the datasets when abundant holdout data are available.

### G.3 Ablation over Number of Online Samples

Here we examine how different algorithms perform as the number of online samples varies. We introduce an additional baseline **BBSE**, which simply uses the label marginal estimate provided by black box shift estimator to reweight the predictions of classifiers. Figure 3 shows an interesting trend

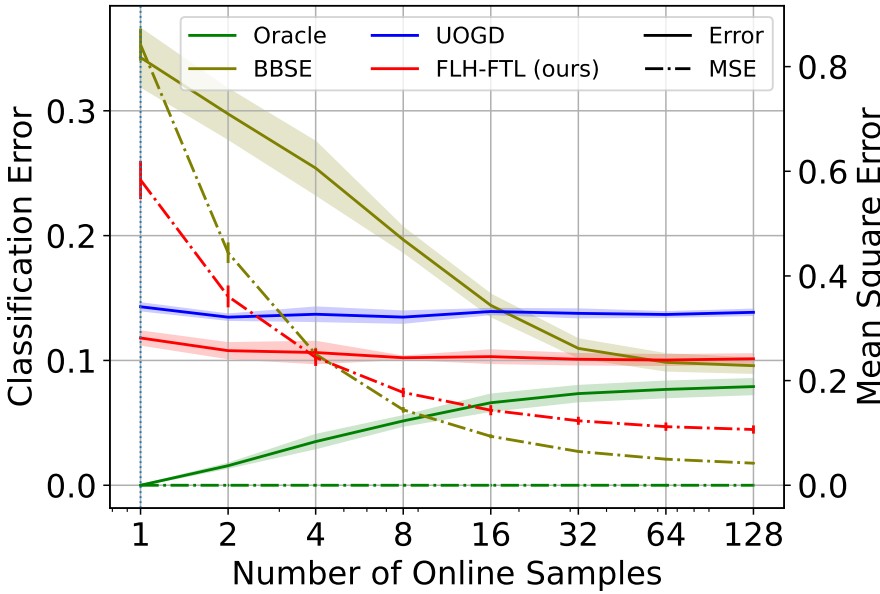

Figure 3: *Performances of online learning algorithms with different number of online samples.* CIFAR-10 results with bernouli shift and limited holdout data. Solid line is the classification error (Error) and the dotted line is the marginal estimation mean squared error (MSE).

that as number of online samples increases, the simple baseline BBSE becomes more competitive and eventually outperforms UOGD, whereas our algorithm remains competitive.

### G.4 Ablation over Types of Marginal Estimates

All the algorithms examined in this work use black box shift estimate (BBSE) [51] to obtain an unbiased estimate of the target label marginal. However, two alternative methods exist: Maximum Likelihood Label Shift (MLLS) [61] and Regularized Learning under Label Shift (RLLS) [2]. Table 9 presents additional results using these two estimates. The results shows using the alternative estimates do not substantially change the performances of the algorithms considered in this work.

| Datasets | | Synthetic | | | CIFAR | | | Fashion | | |
|---|---|---|---|---|---|---|---|---|---|---|
| **MARG EST.** | | BBSE | MLLS | RLLS | BBSE | MLLS | RLLS | BBSE | MLLS | RLLS |
| Bernouli | Base | $8.6_{\pm0.2}$ | $8.6_{\pm0.2}$ | $8.6_{\pm0.2}$ | $16_{\pm0}$ | $16_{\pm0}$ | $16_{\pm0}$ | $15_{\pm0}$ | $15_{\pm0}$ | $15_{\pm0}$ |
| | OFC | $6.4_{\pm0.6}$ | $6.4_{\pm0.6}$ | $6.4_{\pm0.6}$ | $12_{\pm1}$ | $12_{\pm1}$ | $12_{\pm1}$ | $7.9_{\pm0.1}$ | $7.9_{\pm0.1}$ | $7.9_{\pm0.1}$ |
| | FTH | $6.5_{\pm0.6}$ | $6.5_{\pm0.7}$ | $6.5_{\pm0.7}$ | $11_{\pm0}$ | $11_{\pm1}$ | $11_{\pm0}$ | $8.5_{\pm0.3}$ | $8.0_{\pm0.0}$ | $8.6_{\pm0.2}$ |
| | FTFWH | $6.6_{\pm0.5}$ | $6.7_{\pm0.5}$ | $6.6_{\pm0.5}$ | $11_{\pm1}$ | $11_{\pm1}$ | $11_{\pm1}$ | $8.2_{\pm0.6}$ | $7.9_{\pm0.2}$ | $8.3_{\pm0.6}$ |
| | ROGD | $7.9_{\pm0.3}$ | $7.9_{\pm0.3}$ | $7.9_{\pm0.2}$ | $16_{\pm3}$ | $16_{\pm3}$ | $15_{\pm2}$ | $10_{\pm1}$ | $10_{\pm1}$ | $9.6_{\pm1.2}$ |
| | UOGD | $8.1_{\pm0.6}$ | $8.0_{\pm1.0}$ | $8.0_{\pm1.0}$ | $14_{\pm0}$ | $13_{\pm0}$ | $14_{\pm0}$ | $11_{\pm2}$ | $10_{\pm1}$ | $11_{\pm1}$ |
| | ATLAS | $8.0_{\pm1.0}$ | $7.9_{\pm0.5}$ | $8.0_{\pm1.0}$ | $13_{\pm0}$ | $13_{\pm0}$ | $13_{\pm0}$ | $12_{\pm2}$ | $11_{\pm1}$ | $12_{\pm1}$ |
| | FLH-FTL (ours) | $\mathbf{5.4_{\pm0.7}}$ | $\mathbf{5.4_{\pm0.8}}$ | $\mathbf{5.4_{\pm0.7}}$ | $\mathbf{10_{\pm0}}$ | $\mathbf{10_{\pm1}}$ | $\mathbf{10_{\pm0}}$ | $\mathbf{7.7_{\pm0.4}}$ | $\mathbf{7.3_{\pm0.3}}$ | $\mathbf{7.6_{\pm0.3}}$ |
| Sinusoidal | Base | $8.2_{\pm0.3}$ | $8.2_{\pm0.3}$ | $8.2_{\pm0.3}$ | $16_{\pm0}$ | $16_{\pm0}$ | $16_{\pm0}$ | $15_{\pm0}$ | $15_{\pm0}$ | $15_{\pm0}$ |
| | OFC | $5.5_{\pm0.2}$ | $5.5_{\pm0.2}$ | $5.5_{\pm0.2}$ | $11_{\pm0}$ | $11_{\pm0}$ | $11_{\pm0}$ | $7.1_{\pm0.1}$ | $7.1_{\pm0.1}$ | $7.1_{\pm0.1}$ |
| | FTH | $5.7_{\pm0.3}$ | $5.7_{\pm0.2}$ | $5.7_{\pm0.2}$ | $\mathbf{11_{\pm0}}$ | $11_{\pm0}$ | $\mathbf{11_{\pm0}}$ | $6.9_{\pm0.4}$ | $6.6_{\pm0.2}$ | $6.8_{\pm0.4}$ |
| | FTFWH | $5.7_{\pm0.3}$ | $5.6_{\pm0.2}$ | $5.7_{\pm0.3}$ | $\mathbf{11_{\pm0}}$ | $11_{\pm0}$ | $\mathbf{11_{\pm0}}$ | $6.9_{\pm0.4}$ | $6.6_{\pm0.3}$ | $6.9_{\pm0.4}$ |
| | ROGD | $7.2_{\pm0.6}$ | $7.2_{\pm0.6}$ | $7.2_{\pm0.6}$ | $13_{\pm0}$ | $13_{\pm0}$ | $13_{\pm0}$ | $8.2_{\pm0.7}$ | $8.9_{\pm0.6}$ | $8.2_{\pm0.3}$ |
| | UOGD | $7.5_{\pm0.6}$ | $7.4_{\pm0.5}$ | $7.4_{\pm0.5}$ | $14_{\pm1}$ | $13_{\pm1}$ | $14_{\pm1}$ | $11_{\pm2}$ | $9.4_{\pm0.9}$ | $11_{\pm2}$ |
| | ATLAS | $7.5_{\pm0.6}$ | $7.4_{\pm0.6}$ | $7.4_{\pm0.6}$ | $13_{\pm1}$ | $13_{\pm1}$ | $13_{\pm1}$ | $12_{\pm2}$ | $11_{\pm1}$ | $12_{\pm2}$ |
| | FLH-FTL (ours) | $\mathbf{5.4_{\pm0.4}}$ | $\mathbf{5.4_{\pm0.3}}$ | $\mathbf{5.4_{\pm0.4}}$ | $\mathbf{11_{\pm0}}$ | $\mathbf{10_{\pm0}}$ | $\mathbf{11_{\pm0}}$ | $7.0_{\pm0.0}$ | $6.6_{\pm0.2}$ | $6.9_{\pm0.4}$ |

Table 9: *Performances of online learning algorithms with different types of marginal estimates with low amount of holdout data.*

### G.5 Additional results and details on the SHL dataset

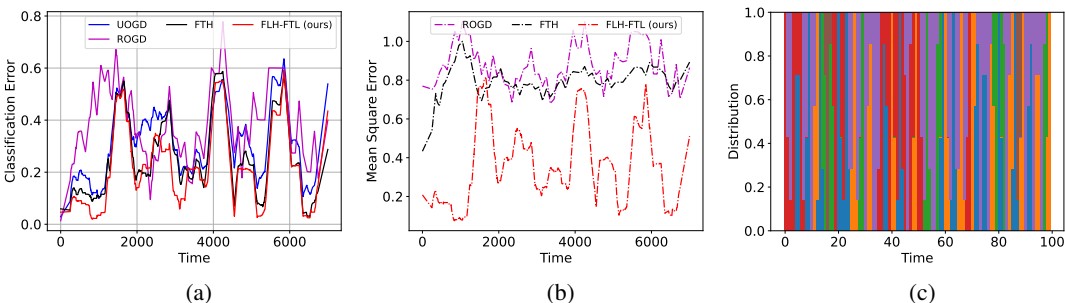

| | | |
|---|---|---|
| (a) | (b) | (c) |

Figure 4: *Additional results and details on the SHL datasets with real shift.* **(a) and (b):** The accuracies and mean square errors in label marginal estimation on SHL dataset over 7,000 time steps with limited amount of holdout data. **(c):** Label marginals of the six classes of SHL dataset. Each time step here shows the marginals over 700 samples.

### G.6 Reweighting Versus Retraining Linear Layer

Here we compare the efficacies of re-weighting (RW-FLH-FTL) and retraining (RT-FLH-FTL) given the marginal estimate provided by FLH-FTL; the latter retrains the last linear layer on the loss of the holdout data re-weighted by the marginal estimate. Note that RW-FLH-FTL corresponds to FLH-FTL in the main text. We retrain RT-FLH-FTL for 50 epochs at each time step. To compare against the best possible retrained classifiers, we used all the holdout data for retraining. Table 10 shows that retraining is often worse and at best similar to re-weighting in performance, despite greater computational cost and need for holdout data.

| Datasets | Synthetic | | | | CIFAR | | | |
|---|---|---|---|---|---|---|---|---|
| Shift | Mon | Sin | Ber | Squ | Mon | Sin | Ber | Squ |
| Base | $8.7_{\pm 0.1}$ | $8.2_{\pm 0.3}$ | $8.6_{\pm 0.3}$ | $8.5_{\pm 0.2}$ | $17_{\pm 0}$ | $16_{\pm 0}$ | $17_{\pm 0}$ | $17_{\pm 0}$ |
| OFC | $6.8_{\pm 0.1}$ | $5.5_{\pm 0.2}$ | $6.7_{\pm 0.2}$ | $6.7_{\pm 0.3}$ | $14_{\pm 0}$ | $11_{\pm 0}$ | $13_{\pm 0}$ | $13_{\pm 1}$ |
| FTFWH | $7.0_{\pm 0.0}$ | $5.5_{\pm 0.3}$ | $6.8_{\pm 0.2}$ | $6.8_{\pm 0.3}$ | $13_{\pm 0}$ | $10_{\pm 0}$ | $12_{\pm 0}$ | $13_{\pm 0}$ |
| UOGD | $7.4_{\pm 0.1}$ | $7.0_{\pm 0.0}$ | $7.6_{\pm 0.4}$ | $7.6_{\pm 0.2}$ | $\mathbf{12}_{\pm 0}$ | $\mathbf{10}_{\pm 0}$ | $\mathbf{11}_{\pm 0}$ | $13_{\pm 0}$ |
| RW-FLH-FTL (ours) | $\mathbf{6.3}_{\pm 0.2}$ | $\mathbf{5.3}_{\pm 0.2}$ | $\mathbf{5.4}_{\pm 0.3}$ | $\mathbf{5.5}_{\pm 0.2}$ | $\mathbf{12}_{\pm 0}$ | $\mathbf{10}_{\pm 0}$ | $\mathbf{11}_{\pm 0}$ | $\mathbf{12}_{\pm 0}$ |
| RT-FLH-FTL (ours) | $6.7_{\pm 0.1}$ | $6.2_{\pm 0.2}$ | $6.0_{\pm 0.0}$ | $6.3_{\pm 0.4}$ | $\mathbf{12}_{\pm 0}$ | $\mathbf{10}_{\pm 0}$ | $\mathbf{11}_{\pm 0}$ | $\mathbf{12}_{\pm 0}$ |

Table 10: *Comparison of performances of re-weighting and retraining strategies with high amount of holdout data.*

