# OpenReview forum: "Online Label Shift: Optimal Dynamic Regret meets Practical Algorithms"
_NeurIPS.cc/2023/Conference — NeurIPS 2023 spotlight_

### Official Review · Reviewer_vA3S · 2023-06-28

**Soundness:** 4 excellent
**Presentation:** 4 excellent
**Contribution:** 4 excellent
**Rating:** 8
**Confidence:** 3

**Summary:**

The general scope is online classifier learning under online label shift where the key assumption is that the class conditionals $Q(x|y)$ are invariant. The paper focuses on both unsupervised and supervised problems, which they coined as Unsupervised Online Label Shift (UOLS) and Supervised Online Label Shift (SOLS).

Key previous works in the UOLS problem [76, 8] setup a regret function and adapt the initial classifier via minimising the regret function over time.  [76] optimises the classifier via reweighting the predicted distribution $\hat{q}_t$. [8] retrains the classifier at every step. However, both suffer from convexity issues of the losses. This paper integrates the best of both [76] and [8] in a single framework. On top of that they avoid any previous convexity issue by reducing the online label shift problem to the problem of online regression of the class marginal distribution function. Tight theoretical bounds under the usual Lipschitz condition are provided.

The second part of the paper is an extension of the approach from the UOLS problem to the SOLS problem. The main contribution here is that theoretical bounds in (weighted) Empirical Risk Minimization are generalised to the SOLS context. Other contributions involve in making online model training faster without compromising the theoretical bounds too much.

Finally, there is an extensive empirical study based on well known datasets. However, in many of the datasets used for the experiments, the label shift dynamics are simulated.

**Strengths:**

The paper is very well written and very detailed. The authors were very clear of their contributions and limitations of the proposed approaches.

In my view, the key novelty of the paper is at reformulating the online label shift problem as the problem of online regression of the marginal distribution function, which previous works have overlooked. Everything else in the paper develops from that idea in naturally. This key novelty is clever because the new problem can be solved efficiently with existing tools.

The provided bounds appear to be quite tight in terms of complexity, although I am not sure whether in practice the coefficient of the highest order term plays a significant role. Proofs in the appendices are simple applications of probabilistic tools, concentration theories and a few key results in online learning theories. I was able to follow without any major trouble.

The finding in appendix G.6 is interesting, which provides an evidence that retraining does not work better than reweighting, despite an additional computational cost for retraining.

**Weaknesses:**

Although well acknowledged in the paper, the fact that the approach, like [8], requires memory that scales linearly with time is quite a concern from a practical point of view. It would be good to have some quantification results regarding memory usage in the experiments.

Although the paper addresses online label shift, among 6 datasets used in the experiments for UOLS and SOLS, 5 of them use simulated label shift data. Only SHL contains real-world label shift data.

I understand that the scope of the paper is online label shift where $Q_t(x|y)$ is unchanged over time. Still I find this scope is rather restrictive compared to real-world data. In the data collection business that I have been involved, data are usually collected from many sources, with large differences in collection periods and frequencies of data being collected per source. With external factors like that it is hard to assume that $Q_t(x|y)$ is unchanged over time. The SHL dataset used in section 5 seems to be the only dataset in which the label shift part of the data is real. However, given that it was collected by only 3 participants, it is hard to justify whether the dataset reflects what happens in the real world.

Perhaps the authors could try with some recent consumer-related datasets (e.g. clothes, food) that contain both pre-Covid and post-Covid periods? During the Covid-19 pandemic, so many shops closed, leading to a large-scale gradual decrement of data that can be collected. Since 2022, many shops have reopened, leading to a gradual increment of collected data.

Maybe a typo. However, on line 112, why does $\delta$ not play a role in the big $\tilde{O}$ function?



**Questions:**

I do not have any major question. The paper is very good to me, except for a few practical concerns raised in the Weaknesses section.

**Limitations:**

I do not see any potential negative societal impact of the work.

---

> ### Author Rebuttal · Authors · 2023-08-09
>
> We thank the reviewer for your generous appraisal of the main theoretical idea. We are glad that you find the paper detailed and well-written.
>
> >**Although well acknowledged in the paper, the fact that the approach, like [8], requires memory that scales linearly with time is quite a concern from a practical point of view. It would be good to have some quantification results regarding memory usage in the experiments**
>
> We will add a table detailing the memory requirement in the appendix.
>
> >**Although the paper addresses online label shift, among 6 datasets used in the experiments for UOLS and SOLS, 5 of them use simulated label shift data. Only SHL contains real-world label shift data… The SHL dataset used in section 5 seems to be the only dataset in which the label shift part of the data is real… Perhaps the authors could try with some recent consumer-related datasets**
>
> We thank the reviewer for suggesting consumer-related datasets as an interesting potential benchmark for our problem. Indeed, curating more real datasets where the label shift assumption is approximately met would be interesting. We are looking into suitable datasets, and will leave curation of these relevant datasets to subsequent works.
>
> >**Maybe a typo. However, on line 112, why does $\delta$ not play a role in the $\tilde O$ function?**
>
> Factors of $\log(1/\delta)$ are subsumed in $\tilde O$ as described in Line 90. We can remind the reader in-place in the updated draft.

---

> > ### Comment · Reviewer_vA3S · 2023-08-14
> > **Thank you!**
> >
> > Thank you for your reply. I'd keep my current ratings.

---

### Official Review · Reviewer_hAtb · 2023-07-04

**Soundness:** 4 excellent
**Presentation:** 4 excellent
**Contribution:** 4 excellent
**Rating:** 7
**Confidence:** 3

**Summary:**

The authors study the important problem of learning in an online setting.
In particular, they focus on two important problems:
- Online Unsupervised learning with Label Shift (UOLS)
- Online Supervised Learning with Label shift (SOLS)
In both cases, they assume, as in previous work, that the distribution of the samples can change at each step, however, the marginal distribution of the covariate X  given the label Y is the same. Their algorithm is adaptive and does not require knowledge of the distribution drift. They show that their results on dynamic regret are (almost) optimal in a minimax sense. Unlike previous work, they do not require a convexity assumption on the family of the losses, which is an important technical contribution. They corroborate their theoretical findings with experiments.





**Strengths:**

Main strengths:

- The paper is very well written, easy to follow, and the results are sound. The assumptions used are discussed and well-motivated.

- They provide a major technical contribution in dropping the convexity assumption of the previous work.

- They show the (almost) tightness of their results in a minimax sense.

- They show an interesting algorithm to reduce the number of training steps in the SOLS setting so that they do not need to train a new model at each step. This is a very interesting result, as this is a common problem of many previous works of learning with distribution drift.

- Their theoretical findings are corroborated by extensive experiments. Moreover, I appreciate the effort of the authors to address many problems related to how to obtain the optimal minimax regret efficiently in practice. (Appendix D).




**Weaknesses:**

The results are limited to changes in the probability of obtaining an element of a certain class (label shift). This is only a specific setting of learning with distribution drift. Their technique looks limited to this specific setting, as it revolves around estimating (regression) the class probabilities. It does not seem straightforward to extend this work or use the results of this work for the more general problem of learning with generic distribution drift.

The experiments are only run with respect to data where synthetic drift is introduced.

**Questions:**

-  With respect to Remark 10: The black box algorithm of appendix D can be used to improve the running time, as it detects intervals in which there is little change and we do not need to train again.
By using a similar argument, is it possible to say that if there is a big change in the label probabilities, then the older data is significantly less useful, and we can stop storing it?

**Limitations:**

The authors discuss limitations, assumptions, and possible future work.

---

> ### Author Rebuttal · Authors · 2023-08-09
>
> We thank the reviewer for your thoughtful feedback. We are glad that you find the paper well-written, and are grateful that you appreciate the experimental and theoretical results.
>
> >**The results are limited to changes in the probability of obtaining an element of a certain class (label shift). This is only a specific setting of learning with distribution drift. Their technique looks limited to this specific setting, as it revolves around estimating (regression) the class probabilities. It does not seem straightforward to extend this work or use the results of this work for the more general problem of learning with generic distribution drift.**
>
> As the reviewer correctly points out, our methods are applicable to the setting of label shift. We documented this in Appendix A (Limitations section). However, we note that absent causal assumptions on the distribution shift, it is in general impossible to do unsupervised domain adaptation. It is indeed an interesting future direction to extend our work to other types of shifts we see in an online setting. Our work only takes the first steps towards this bigger goal.
>
> >**The experiments are only run with respect to data where synthetic drift is introduced.**
>
> In fact, we provide experimental results on the SHL datasets in the appendix where the shift is natural. We believe that this demonstrates the superiority of our method over other alternatives even under real shift.
>
> >**With respect to Remark 10: The black box algorithm of appendix D can be used to improve the running time, as it detects intervals in which there is little change and we do not need to train again. By using a similar argument, is it possible to say that if there is a big change in the label probabilities, then the older data is significantly less useful, and we can stop storing it?**
>
> As the reviewer mentioned, if there is a big change in the label probabilities, then the older data is significantly less useful. However, there is still a possibility of older data becoming more relevant at a later point of time. Thus to prevent catastrophic forgetting we keep the old data around.

---

> > ### Comment · Reviewer_hAtb · 2023-08-18
> >
> > I would like to thank the authors for their response. After also reading the other reviews, I would like to keep my rating. I think this is a good paper that should be accepted.

---

### Official Review · Reviewer_Gcxc · 2023-07-05

**Soundness:** 3 good
**Presentation:** 3 good
**Contribution:** 3 good
**Rating:** 8
**Confidence:** 3

**Summary:**

this paper applies techniques from online regression oracles to tackle the problems of unsupervised and supervised label shift, where the marginal probabilities of the labels can be changed over time by an adversary.

**Strengths:**

the main contributions include:
C1 an algorithm that adapts to the changing labels without knowing the parameter V_T
C2 an algorithm whose online learning assumptions are verified
C3 an algorithm compatible with black box access to the learner, to enable deep neural networks and decision trees to be used as learners
experiments illustrate the proposed algorithm is promising

the paper is well written and extensive experiments are performed with many baselines

**Weaknesses:**

some minor issues:

1 figure 2 does not explain what the uncertainty bands are. I assume these are std's or are they standard errors? similairly for the tables. figure 2c misses erroprbars or are they not visible?

2 the appendix is extremely long, I did not have time to go over all the details, and I am inclined to suggest a full paper would be better reviewed at a journal. I thought its a shame that fig. 3 (appendix) which shows interesting trends is not included in the main body.

**Questions:**

3 the lowerbounds do seem to be of a slightly different form than the upperbounds. does this matter? why (not)? why is it essential to assume V_T is smaller than T*c for some c for the lowerbounds?

---

> ### Author Rebuttal · Authors · 2023-08-09
>
> We want to thank the reviewer for your positive feedback and for championing our paper. We are glad that you find the paper well-written and appreciate the experiments.
>
> >**1 figure 2 does not explain what the uncertainty bands are. I assume these are std's or are they standard errors? similairly for the tables. figure 2c misses erroprbars or are they not visible?**
>
> Indeed, the uncertainty bands are standard deviations. We thank the reviewer for bringing the potential confusion to our attention, and we will clarify this in the updated draft. We decided to leave out the error bars on figure 2c for visual clarity as they are insignificant compared to the advantage in our method.
>
> >**I thought its a shame that fig. 3 (appendix) which shows interesting trends is not included in the main body.**
>
> We thank the reviewer for appreciating this finding from the supplementary materials. We had to make a hard choice given space constraints. If given an extra page for the final version, we will include this figure in the main paper.
>
> >**3 the lowerbounds do seem to be of a slightly different form than the upperbounds. does this matter? why (not)? why is it essential to assume $V_T$ is smaller than $T*c$ for some $c$ for the lowerbounds?**
>
> It is a known result from dynamic regret minimization literature that if the variation $V_T$ of the target marginals is not sublinear in $T$ , one cannot hope to get sublinear regret bounds (see for eg. [1, 2, 3]). The intuition is that when $V_T$ is not sublinear, the target moves too fast so that it is impossible to track the target by an online learner.
>
>
> [1] Lijun Zhang,  Shiyin Lu and  Zhi-Hua Zhou. Adaptive online learning in dynamic  environments. NeurIPS 2018
>
> [2] Dheeraj Baby and Yu-Xiang Wang. Optimal Dynamic Regret in Exp-Concave Online Learning. COLT 2021
>
> [3] Omar Besbes, Yonatan Gur and Assaf Zeevi. Non-stationary Stochastic Optimization. Operations research, 2015

---

### Official Review · Reviewer_Cpae · 2023-07-06

**Soundness:** 3 good
**Presentation:** 2 fair
**Contribution:** 3 good
**Rating:** 7
**Confidence:** 4

**Summary:**

This paper delves into the supervised and unsupervised online label shift problem, which focuses on adapting to changing class marginals using online data. The authors propose new algorithms that convert the adaptation problem into an online regression task, guaranteeing the minimax optimal dynamic regret. These methods leverage online regression oracles to track the shifting proportions. Evaluations of various online label shift scenarios show improved performances of the proposed methods.

**Strengths:**

1. The manuscript is neatly structured and clearly drafted.

2. The authors have comprehensively explained both the problem setup and their proposed solutions.

3. The theoretical foundations elaborated in the manuscript appear to be solid. Importantly, the authors extend the concept of non-stationary online learning to scenarios encompassing non-convex loss functions, albeit under a Lipschitzness assumption. This extension significantly broadens the applicability of non-stationary online learning, notably in the context of neural networks or other sophisticated prediction models.

**Weaknesses:**

While I am mainly positive about the paper. However, there remain some issues that may need to be further addressed for a more compelling submission:

1. The usage of the term "sample efficient" in both the abstract and experimental sections appears confusing to me. Sample efficiency is typically understood in a theoretical context. However, the proposed method does not theoretically demonstrate sample efficiency (consistent with previous methods). Moreover, the experiments merely evaluate the size of the hold-out offline labeled data, hardly serving as evidence for "sample efficiency." Consequently, the paper does not sufficiently convince me of this.
2. It is commendable that, unlike its predecessors, the proposed method does not hinge on the convexity of the functions (line 37). Nevertheless, given the importance of the Lipschitz constant in this paper, questions arise regarding the control of the Lipschitz constant G in complex models such as Deep Neural Networks (DNNs) and decision trees. For instance, decision trees, lacking continuity, presumably possess an infinite G.
3. In Theorem 6 (line 834), a simple union bound is utilized to derive the lower bound of the online label shift problem, resulting in a term $|\mathcal{H}|$ in the lower bound. Yet, can this bound accommodate complex scenarios where $\mathcal{H}$ is the DNNs or decision trees, given that $|\mathcal{H}|$ would be infinite in such instances?
4. Based on the points raised in Weaknesses 1, 2, and 3, the paper's title, "Online Label Shift: Optimal Dynamic Regret with Non-convex Implementation", appears somewhat overclaimed to me:
    * In Remark 10, the authors acknowledge that Algorithm 4 necessitates storing and reprocessing all past data in each iteration, which contradicts the fundamental principles of online learning. Consequently, the term "Practical Algorithms" in the title seems overclaimed to me.
    * A more fitting title, such as "Online Label Shift: Optimal Dynamic Regret with Non-convex Implementation," could be considered, given the paper's principal contribution lies in addressing non-convexity. Such a title could potentially draw more attention from readers.
5. The authors assert (line 55) that their approach achieves the best-of-both worlds of previous works. However, the meaning of "worlds" is unclear. Could the authors elucidate this further?
6. The theoretical exposition is presented in a complicated manner and is rather daunting to read. It would potentially be more friendly to a broader audience if the authors simplified and condensed the theoretical content.
7. Within the methodology section, the discrepancy between estimated and actual label marginals is converted into a 2-norm gap, which subsequent methods seek to optimize directly. However, in the context of label shift problems, is this direct optimization of the 2-norm (square loss) between distributions appropriate? Is the 2-norm excessively susceptible to out-of-distribution (OOD) data? Perhaps, alternative measures like KL-divergence or cross-entropy could be more suitable?
8. Some other relevant works related to online distribution shift should be considered to be explored and discussed, such as:

    * Kumar et al., Understanding Self-Training for Gradual Domain Adaptation;
    * Zhang et al., Adapting to Continuous Covariate Shift via Online Density Ratio Estimation;
    * Zhou et al., Online Continual Adaptation with Active Self-Training.
9. I wonder what are the "NA"s mean in the experimental tables?
10. I am unsure about the FLH and Alligator algorithms (line 193). Could the authors briefly introduce these algorithms and explain their function?
11. line 196: The averaging within "intervals" xxx, can the author detail explain what the "intervals" is? and how to divide these intervals.
12. The paper can be further refined to enhance readability, and there are several typographical errors:
     * line 31: Wu et al. [76] only controls -> Wu et al. [76] only control
     * line 35: However their approach based on -> However, their approach is based on
     * line 82: the extend of distribution drift -> the extent of distribution drift.
     * line 94: i.e. -> i.e.,
     * line 96: respectively -> , respectively
     * line 314: i.e. -> i.e.,
     * line 1055: $\alpha_t = 1-\alpha_{t-1}$.
     * line 1079: to better performance -> to a better performance

**Questions:**

The paper is mainly sound and well-organized, but there remain some concerns for me, see the Weaknesses part above.

**Limitations:**

As delineated in the Weakness section, the authors could provide a more detailed elucidation of the methodologies employed, and potentially consider revising the title to garner a broader impact. I am open to adjusting the evaluation score if the authors address my concerns.

---

> ### Author Rebuttal · Authors · 2023-08-09
>
> We are thankful for Reviewer Cpae’s constructive feedback and positive perspective on the work. We have incorporated your feedback, and believe they have greatly enhanced the manuscript.
>
> >**... "sample efficient"...**
>
> We apologize for the confusion. The term “sample efficient” in the theoretical context of online learning means that the regret upper bounds of our algorithms match the information-theoretic lower bounds. In experimental context, we’re referring to the finding that in the low amount of data regime (on the left of Figure 2 a/b), our method performs significantly better than prior works, and thus is more efficient in the number of labeled samples required. We’ll clarify this in the introduction.
>
> >**… Lipschitz constant G…**
>
> We would like to clarify one confusion about the Lipschitzness. We note that the Lipschitzness of the losses are required only wrt the re-weighting measure $p \in \Delta_K$ (see Assumption 2) and *not* wrt the covariate $x$. The effect of the covariate space is marginalized away via expectation over the random draw of (x,y) pairs (ref. Lines 173-177). Hence the discontinuities in the decisions of the classifier across the feature space do not blow up the Lipschitz constant wrt re-weighting measure to infinity. We believe that this form of Lipschitzness is milder than imposing smoothness constraints on the loss across the feature space.
>
> >**In Theorem 6 ...**
>
> While we prove the above result for finite hypothesis sets, the extension to infinite sets is straight-forward by standard covering net arguments [70]. In doing so, the terms of the form $\log |\mathcal{H}|$ in Theorem 6 will be replaced by the Rademacher complexity of the hypothesis class. In fact, we initially had a remark on this, but removed it due to space constraints. We will add this remark in the final version.
>
> >**... more fitting title…**
>
> We apologize if the title is misleading. The “practical algorithms” refers only to the algorithms used in the unsupervised setting. We will clarify this in the introduction. We are also happy to consider changing the title in the final version.
>
> >**... best-of-both worlds…**
>
> The phrase best-of-both worlds used in line 55 foreshadows the arguments in Section 3, especially lines 144-148.
>
> We briefly describe it here. The work of [76] allows us to use expressive initial classifiers such as DNN while only controlling the *static* regret against a fixed classifier (though their theoretical guarantees hinge on convexity of losses). The work of [8] allows to control for dynamic regret against a sequence of classifiers while imposing restrictive *convexity* assumptions on the losses. The notion of dynamic regret is more adequate than static regret particularly in the face of distribution shifts.  Our work combines the best features of both of the above works by controlling for dynamic regret while not imposing restrictive convexity assumptions thereby allowing us to exploit the power of highly expressive classifiers.
>
> >**... optimization of the 2-norm …**
>
> Indeed, we may consider alternative losses like KL-divergence for tracking the marginals. In fact, if we use KL-divergence instead, we would arrive at similar guarantees as in Theorem 2. This is due to the fact that KL loss belongs to the family of exp-concave functions [12]. The  regret rates for KL and squared loss regression are the same (see [4]). Further, online regression with KL loss involves inverting a matrix, which will result in increased run-time in comparison to squared loss regression. Due to these reasons we didn’t pursue the direction of KL-loss. On the other hand, the minimax regret guarantees of Theorems 2 and 5 indicate that squared loss is sufficient to get optimal rates.
>
> >**... other relevant works …**
>
> We thank the reviewer for bringing relevant works to our attention. We will include these references in the updated draft.
>
> >**... "NA"s …?**
>
> NA stands for "Not Applicable," indicating that base classifiers don't use label marginal estimates for reweighting predictions. Thus, the mean square error in label marginal estimate (the row MSE) is not applicable to these classifiers. We thank the reviewer for pointing out this potential confusion, and will clarify it in the updated draft.
>
> >**... FLH and Alligator …**
>
> Both algorithms [7, 38] can be used as online regression algorithms in the sense defined in Definition 1. At a high level, both are basically ensemble methods. They maintain a pool of online averaging base learners. Each base learner only averages a subset of past observations in the rounds {1,2,...,T}. At any round the base learners submit their predictions to a meta-algorithm, which then combines the individual predictions via the ensemble method of Exponetiated Weighted Averages.
>
> A primary challenge in the online regression problem is to decide how much of the past data we must average at each time step. FLH or Aligator optimally and adaptively average over just the right amount of past data points.
>
> >**... the "intervals" …**
>
> By intervals, we are referring to the intervals in which the ground truth label marginals drift sufficiently slowly. Such intervals are only known to an unrealistic oracle. As the reviewer points out, we do not have such oracle information about the low drift intervals in practice. The remarkable property of online regression algorithms is that they can incur a total estimation error that is closely comparable to the error that would have been made by an ideal estimator with the aforementioned oracle knowledge. We refer the reviewer to [4] on more formal details about such oracle inequalities.
>
> >**... typographical errors**
>
> We thank the reviewer for pointing out these typos and will correct them in the final draft.

---

> > ### Comment · Reviewer_Cpae · 2023-08-15
> >
> > I would like to thank the authors for the detailed feedback. My concerns have been largely addressed, and I am inclined to revise my score in favor of acceptance. I appreciate the depth of analysis and the contributions made in this paper. However, I'd like to suggest the authors to further refine the presentation to enhance the accessibility to a broader audience. In particular, incorporating more details on the algorithms and supplementing necessary explanations,  especially for the theoretical results, would be greatly beneficial for the reader in grasping the main ideas. Additionally, a more descriptive title reflecting the paper's core contributions could further enhance its impact. I hope the authors find these suggestions beneficial.
> >
> > A minor note: In the context of online learning/optimization, wouldn't terms like "optimal" or "tight" be more common than "sample efficient" when discussing the regret?

---

> > > ### Author Response · Authors · 2023-08-16
> > > **Response to Reviewer Cpae's official comment**
> > >
> > > We thank the reviewer for their response and for increasing their score. We will work on the updated version to include the readability for a broader audience. In particular, we will include an overview of the theory section at the beginning of Section 3 and Section 4 which explains and distills our main findings.
> > >
> > > Regarding your note about using terms like "tight" and "optimal", we agree with you that these terms have also been used popularly in the literature. The term "sample efficient" is also synonymously used. We explain the reasoning behind it as follows:
> > >
> > > The notion of regret in online learning is intimately tied to the notion of excess risk in statistical learning theory as introduced by Vapnik (for eg. see https://www.mit.edu/~rakhlin/courses/stat928/stat928_notes.pdf). If we divide our regret bounds by horizon $T$, we get that the excess risk against the comparator is bounded by $O((V_T/T)^{2/3} + T^{-1/2})$, where risk is defined as the average loss incurred in $T$ rounds. This bound translates equivalently to a sample complexity bound, indicating the rounds needed for an online learner to achieve an excess risk of $\epsilon$. Our algorithms attain the information-theoretically minimal sample complexity, thereby making them inherently sample efficient.

---

### Author Rebuttal · Authors · 2023-08-09

We are thankful to the reviewers for their careful consideration of our work and thoughtful feedback. We are glad to find that the reviewers appreciate our work and gave positive scores (8, 8, 7, 4). We also appreciate that all reviewers find the paper well-written. While reviewer Cpae has some clarification questions and concerns, we believe that they are addressable in this round of rebuttal. We have adjusted the manuscript as suggested, and believe the changes has improved the exposition. Please let us know if there are any remaining concerns. We will be happy to answer any further questions you may have.

---

### Decision · Program_Chairs · 2023-09-21

**Decision:**

Accept (spotlight)

**Comment:**

In this paper, the authors investigate online label shift under supervised and unsupervised settings. The goal is to develop algorithms that adapts to the non-stationarity in the label drift. Under the Lipschitzness condition, the authors reduce the adaptation problem to online regression, and then leverage existing oracles to establish minimax optimal dynamic regret rates.

This paper is easy to follow, and the algorithms are well-motivated. Compared to prior work, the authors get rid of the convexity assumption, which is a significant advantage. By doing so, they are able to leverage advanced deep models to achieve superior performance. On the other hand, the authors should further polish their paper based on the suggestions from the reviewers.